# Piecewise Constant Spectral Graph Neural Network

**Vahan Martirosyan**                                              *vahan.martirosyan@centralesupelec.fr*
*Université Paris-Saclay*
*CentraleSupélec, Inria, France*

**Jhony H. Giraldo**                                                      *jhony.giraldo@telecom-paris.fr*
*LTCI, Télécom Paris*
*Institut Polytechnique de Paris, France*

**Fragkiskos D. Malliaros**                                    *fragkiskos.malliaros@centralesupelec.fr*
*Université Paris-Saclay*
*CentraleSupélec, Inria, France*

**Reviewed on OpenReview:** *https://openreview.net/forum?id=sTdVnDWOHX*

## Abstract

Graph Neural Networks (GNNs) have achieved significant success across various domains by leveraging graph structures in data. Existing spectral GNNs, which use low-degree polynomial filters to capture graph spectral properties, may not fully identify the graph's spectral characteristics because of the polynomial's small degree. However, increasing the polynomial degree is computationally expensive and, beyond certain thresholds, leads to performance plateaus or degradation. In this paper, we introduce the **Pie**cewise **Co**nstant Spectral Graph **N**eural Network (PieCoN) to address these challenges. PieCoN combines constant spectral filters with polynomial filters to provide a more flexible way to leverage the graph structure. By adaptively partitioning the spectrum into intervals, our approach increases the range of spectral properties that can be effectively learned. Experiments on nine benchmark datasets, including both homophilic and heterophilic graphs, demonstrate that PieCoN is particularly effective on heterophilic datasets, highlighting its potential for a wide range of applications. The implementation of PieCoN is available at `https://github.com/vmart20/PieCoN`.

## 1 Introduction

Graph Neural Networks (GNNs) (Wu et al., 2021; Zhou et al., 2020) have demonstrated remarkable performance across various application domains. They have been successfully applied in areas such as social network analysis (Panagopoulos et al., 2023), recommendation systems (Wu et al., 2019; Ying et al., 2018), drug discovery (Jiang et al., 2021; Bongini et al., 2021), and materials modeling (Coley et al., 2019; Duval et al., 2023), where data can naturally be represented as graphs. GNNs can be broadly classified into two types: spatial and spectral GNNs. Spatial GNNs (Kipf & Welling, 2017; Veličković et al., 2018; Chien et al., 2021) use a message passing approach to learn node representations by collecting information from neighboring nodes. This approach allows spatial GNNs to capture local structural information and adapt to varying neighborhood sizes. On the other hand, spectral GNNs (Wang & Zhang, 2022; Defferrard et al., 2016; He et al., 2021; Castro-Correa et al., 2024) use the graph's spectral characteristics, such as the graph Laplacian's eigenvalues and eigenvectors, to transform node features. By leveraging the spectral domain, these models can capture global structural patterns and apply graph convolution operations in the frequency domain.

Many existing spectral GNNs use low-degree polynomial filters, which approximate the filtering functions by applying polynomials to the graph's Laplacian matrix or other graph-shift operators (Wang & Zhang, 2022; Defferrard et al., 2016; He et al., 2021). One disadvantage of these low-degree polynomial filters is that they are continuous and, because of the low degree, may not give enough weight to specific eigenvalues, as the

change between closely spaced eigenvalues cannot vary significantly. This can be problematic, particularly in real-world graphs where certain eigenvalues like *zero*[1] have large multiplicities (Lu et al., 2024; Lim et al., 2023).

In this paper, we propose the **Pie**cewise **Co**nstant Spectral Graph **N**eural Network (PieCoN) to overcome this limitation. Our approach combines constant spectral filters with polynomial filters to better capture the spectral properties of the graph. The constant filters are defined by setting the values in the diagonal eigenvalue matrix to ones within specific intervals and zeros elsewhere, effectively isolating different frequency bands.

By partitioning the spectrum into intervals, PieCoN expands the range of spectral characteristics that can be learned, improving the model's performance. Figure 1 compares the response of a JacobiConv filter (Wang & Zhang, 2022) versus the response of a PieCoN filter across the spectrum. The figure shows how polynomial filters produce a smooth response, while piecewise constant filters can sharply focus on selected intervals, capturing crucial spectral properties that polynomial filters miss. For example, PieCoN can create a sharp drop at eigenvalue 0, which is impossible to achieve with low-degree polynomial filters like JacobiConv. This sharp discontinuity is crucial because eigenvalue 0 often has high multiplicity in real-world graphs (as detailed in Table 1) and contains important structural information. Our theoretical analysis provides important insights about spectral GNNs by establishing error bounds for polynomial spectral filtering and proving that our model is invariant to eigenvector sign flips and basis changes. We validate our approach on standard benchmark datasets, showing improved performance, particularly when handling graphs with multiple zero eigenvalues.

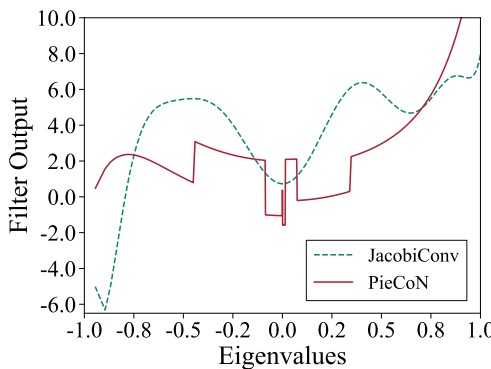

Figure 1: Comparison of JacobiConv and PieCoN trained filters on the Chameleon dataset.

Our main contributions are as follows:

- We introduce a novel spectral GNN model (PieCoN) that uses piecewise constant filters combined with polynomial filters to enhance learning from the spectral properties of graphs.
- We propose a new method to isolate frequency bands in the spectral domain by partitioning the eigenvalue spectrum into intervals, allowing the model to focus on crucial spectral properties.
- We demonstrate, through experiments, that PieCoN outperforms or shows competitive performance against spatial and spectral GNNs on real-world datasets.

## 2  Related Work

GNNs have evolved significantly since the early work by Bruna et al. (2014), who introduced the first modern spectral-based graph convolution network using the graph Fourier transform. Later, Kipf & Welling (2017) simplified this approach with the Graph Convolutional Network (GCN) model, which applies a first-order approximation of spectral filters to make GNNs more scalable. Other important advancements include Graph Attention Networks (GAT) (Veličković et al., 2018), which use attention mechanisms to weigh neighboring nodes differently, and GraphSAGE (Hamilton et al., 2017), which focuses on inductive learning by generating node embeddings through sampling and aggregation techniques.

Message-passing neural networks (MPNNs) (Gilmer et al., 2017) are a class of GNNs where nodes iteratively exchange and aggregate information with their neighbors. Besides GCN and GAT, several other message-passing approaches have emerged to address specific challenges. For instance, the Graph Isomorphism Networks (GIN) model (Xu et al., 2019) is designed to be as powerful as the 1-WL test for distinguishing non-isomorphic graphs, thus improving expressiveness.

---

[1]Note that, for the Laplacian matrix, the multiplicity of eigenvalue zero corresponds to the number of connected components. This correspondence does not hold for the normalized adjacency matrix we employ here.

Spectral GNNs leverage the eigenvalues and eigenvectors of the graph Laplacian to perform graph convolution in the spectral domain. ChebNet (Defferrard et al., 2016) introduced the use of Chebyshev polynomials for spectral filtering, improving the scalability of spectral GNNs. JacobiConv (Wang & Zhang, 2022) further enhances this by employing orthogonal Jacobi polynomials to flexibly learn graph filters. Recent work by Maskey et al. (2023) introduces a novel fractional graph Laplacian approach defined in the singular value domain to address over-smoothing, providing theoretical guarantees for both directed and undirected graphs. For heterophilic graphs with label noise, $R^2LP$ (Cheng et al., 2024) demonstrates that increasing graph homophily can help mitigate the impact of noisy labels.

While previous spectral GNNs have leveraged polynomial filters to approximate spectral properties of graphs, they are limited by their lack of flexibility in handling critical eigenvalues because of the low degree of the polynomial filters. DSF (Guo et al., 2023) addresses this by employing a shared network on positional encodings to learn unique polynomial coefficients per node, highlighting the advantages of node-specific filters over node-unified ones. NFGNN (Zheng et al., 2023) proposes a node-oriented spectral filtering approach that learns specific filters for each node, better adapting to local homophily patterns. PP-GNN (Lingam et al., 2022) also explores piecewise spectral filtering, but our approach differs in several key aspects. While PP-GNN uses a fixed two-part partitioning with low-degree polynomials and continuity constraints, PieCoN implements adaptive $K$-part partitioning with constant filters that allow for discontinuities between intervals. This design enables PieCoN to better capture critical eigenvalues with sharper responses. Additionally, our approach decomposes filters into positive and negative parts (Eq. (10)) and maintains invariance to eigenvector sign flips and basis changes (Proposition 2).

## 3 Background

We consider an undirected graph $G = (\mathcal{V}, \mathcal{E}, \boldsymbol{X})$ with $n$ nodes. Here, $\mathcal{V} = \{v_1, v_2, \ldots, v_n\}$ is the set of nodes, $\mathcal{E} \subseteq \mathcal{V} \times \mathcal{V}$ ($|\mathcal{E}| = m$) represents the set of edges, and $\boldsymbol{X} \in \mathbb{R}^{n \times d}$ is the node feature matrix. The adjacency matrix $\boldsymbol{A} \in \{0,1\}^{n \times n}$ of the graph is defined as $\boldsymbol{A}_{ij} = 1$ if there is an edge between nodes $v_i$ and $v_j$, and $\boldsymbol{A}_{ij} = 0$ otherwise. The degree matrix $\boldsymbol{D} = \mathrm{diag}(d_1, \ldots, d_n)$ is a diagonal matrix with the $i$-th diagonal entry as $d_i = \sum_j \boldsymbol{A}_{ij}$, representing the degree of node $i$. The normalized adjacency matrix $\hat{\boldsymbol{A}}$ is defined as $\hat{\boldsymbol{A}} = \boldsymbol{D}^{-\frac{1}{2}} \boldsymbol{A} \boldsymbol{D}^{-\frac{1}{2}}$. The normalized adjacency matrix is symmetric and can be decomposed as $\hat{\boldsymbol{A}} = \boldsymbol{U} \boldsymbol{\Lambda} \boldsymbol{U}^\top$, where $\boldsymbol{\Lambda}$ is a diagonal matrix containing the eigenvalues $\lambda_i$ of $\hat{\boldsymbol{A}}$, and $\boldsymbol{U}$ is an orthogonal matrix whose columns are the corresponding eigenvectors $\boldsymbol{u}_i$. Let $s$ be the number of distinct eigenvalues of $\hat{\boldsymbol{A}}$, denoted by $\lambda'_1, \lambda'_2, \ldots, \lambda'_s$. For an eigenvalue $\lambda$, we define $\nu(\lambda)$ as the *algebraic multiplicity* of $\lambda$, which is the number of times $\lambda$ appears in the eigenvalues. A graph signal is a function that assigns a scalar value to each node in the graph. Formally, a graph signal can be represented as a vector $\boldsymbol{x} \in \mathbb{R}^n$, where each entry $x_i$ corresponds to the signal value at node $v_i$.

### 3.1 Graph Neural Networks

Graph Neural Networks (GNNs) are deep learning architectures designed to learn from graph-structured data. They can be broadly categorized into two types: spatial GNNs and spectral GNNs. Both approaches aim to learn node representations by leveraging the graph structure and node features but differ fundamentally in how they process graph information.

**Graph Fourier Transform and Spectral GNNs.** The graph Fourier transform (Ortega et al., 2018) of a graph signal $\boldsymbol{x} \in \mathbb{R}^n$ is defined as $\hat{\boldsymbol{x}} = \boldsymbol{U}^\top \boldsymbol{x}$, and its inverse is given by $\boldsymbol{x} = \boldsymbol{U} \hat{\boldsymbol{x}}$ (Shuman et al., 2013). This transform projects a graph signal from the spatial domain (node space) to the spectral domain (frequency space), similar to how the classical Fourier transform operates on time signals.

The spectral filtering of a signal $\boldsymbol{x}$ with a kernel $\boldsymbol{v}$ is defined as:

$$\boldsymbol{z} = \boldsymbol{v} *_G \boldsymbol{x} = \boldsymbol{U} \left( \hat{\boldsymbol{v}} \odot \hat{\boldsymbol{x}} \right) = \boldsymbol{U} \hat{\boldsymbol{V}} \boldsymbol{U}^\top \boldsymbol{x}, \tag{1}$$

where $\hat{\boldsymbol{V}} = \mathrm{diag}(\hat{v}_1, \ldots, \hat{v}_n)$ represents the spectral kernel coefficients and $\odot$ denotes element-wise multiplication. Spectral GNNs leverage this graph Fourier transform to define convolution operations in the

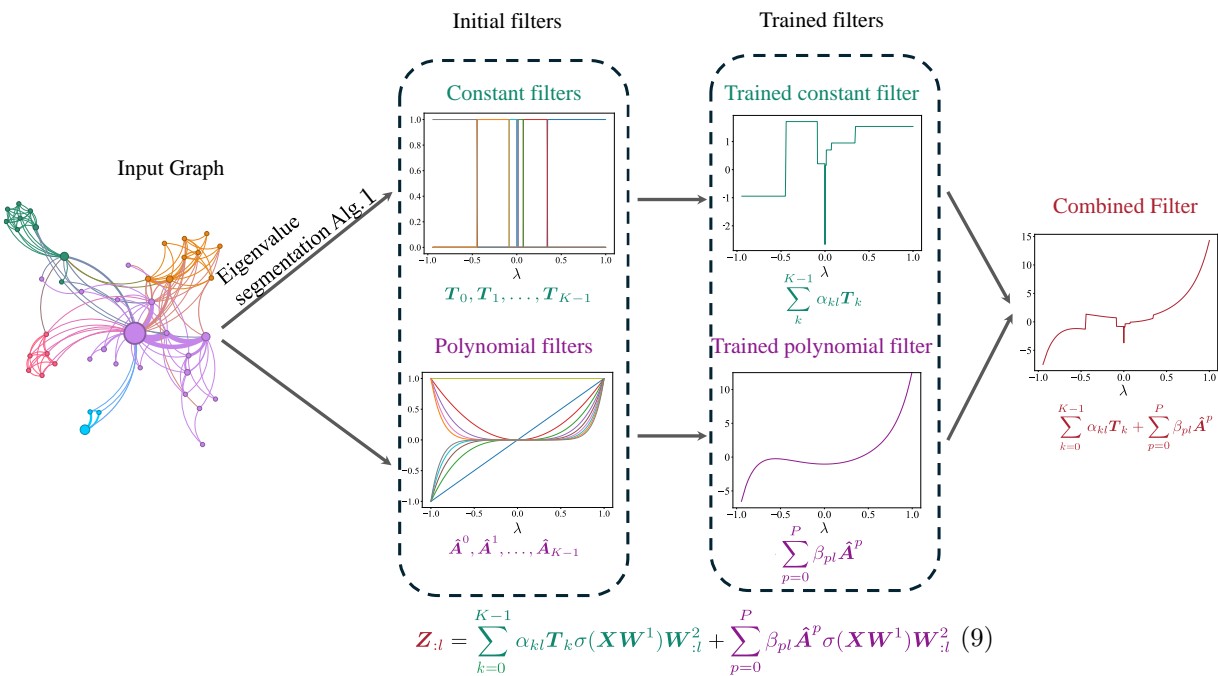

Figure 2: Overview of the PieCoN model. Our method processes an input graph through eigenvalue segmentation (Alg. 1) to create constant filters, while separately applying polynomial filters. These filters are trained and combined to create the final spectral filter.

spectral domain. These models approach graph learning from a signal processing perspective, using the eigendecomposition of graph matrices to define filtering operations.

To avoid the computationally expensive eigen-decomposition, polynomial functions $h(\hat{\boldsymbol{A}})$ are often used to approximate different kernels in spectral GNNs. Specifically, the spectral filter $h(\lambda)$ is parameterized as a polynomial of degree $P$:

$$h(\lambda) = \sum_{p=0}^{P} \beta_p \lambda^p, \tag{2}$$

where $\beta_p$ are learnable coefficients. Consequently, the filtering process can be reformulated as:

$$h(\hat{\boldsymbol{A}})\boldsymbol{x} = \sum_{p=0}^{P} \beta_p \hat{\boldsymbol{A}}^p \boldsymbol{x} = \boldsymbol{U} h(\boldsymbol{\Lambda}) \boldsymbol{U}^\top \boldsymbol{x}, \tag{3}$$

which allows efficient computation of the filtered signal using only matrix multiplications.

**Spatial GNNs.** Spatial GNNs, also known as MPNNs, operate directly on the graph structure by aggregating information from neighboring nodes (Gilmer et al., 2017). The general form of message passing can be expressed as:

$$\mathbf{h}_i^{(l+1)} = \text{UPDATE}\left(\mathbf{h}_i^{(l)}, \text{AGGREGATE}\left(\{\mathbf{h}_j^{(l)} : j \in \mathcal{N}(i)\}\right)\right), \tag{4}$$

where $\mathbf{h}_i^{(l)}$ represents the feature vector of node $i$ at layer $l$, $\mathcal{N}(i)$ is the set of neighboring nodes of $i$, AGGREGATE is a permutation-invariant function that combines information from the neighbors, and UPDATE is a function that updates the node's representation.

Popular examples of MPNNs include the GCN model (Kipf & Welling, 2017), which can be formulated as:

$$\mathbf{H}^{(l+1)} = \sigma\left(\hat{\boldsymbol{A}}\mathbf{H}^{(l)}\mathbf{W}^{(l)}\right), \tag{5}$$

where $\mathbf{H}^{(l)}$ is the matrix of node representations at layer $l$, $\mathbf{W}^{(l)}$ is a learnable weight matrix, and $\sigma(\cdot)$ is a non-linear activation function.

## 4 Piecewise Constant Spectral Graph Neural Network (PieCoN)

Current spectral GNNs often have limited flexibility in how they process graph structures due to their reliance on polynomial filters. We propose PieCoN, a model that combines different types of spectral filters to process graph data in ways that complement the capabilities of polynomial filters. The key steps of our methodology, illustrated in Fig. 2, include: (1) partitioning the graph spectrum into intervals based on significant points identified through spectral analysis, (2) constructing constant spectral filters for each interval to capture global and local spectral properties, and (3) combining constant spectral filters with polynomial filters. Below, we describe the methodology in detail. All the proofs of theorems and propositions are provided in Appendix A.

### 4.1 Identifying Significant Points in the Spectrum

Understanding the spectral properties of a graph is crucial for analyzing its structure. A key challenge is identifying significant points in the eigenvalue spectrum, which can reveal important structural insights (Fig. 3). Algorithm 1 addresses this by identifying large gaps in the eigenvalue spectrum, which can indicate distinct frequency bands in the graph's spectral representation and highlight structural changes or regions of high spectral variation (Von Luxburg, 2007; Fiedler, 1973; Chung, 1997). By adaptively partitioning the eigenvalue space around these critical points, PieCoN can capture the most informative spectral features of the graph.

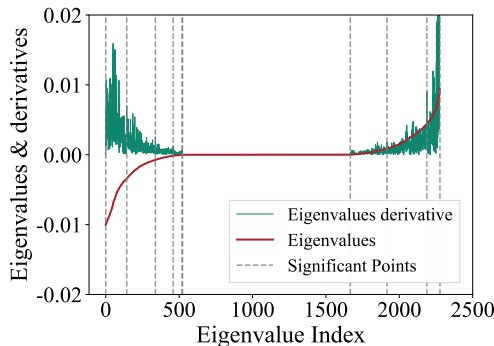

Figure 3: Using the derivative of eigenvalues to identify significant points which show relatively high changes in the spectrum.

Algorithm 1 identifies significant points in the eigenvalue spectrum by analyzing local patterns in the distribution of eigenvalues. For each position $i$ in the sorted sequence of eigenvalues, the algorithm quantifies how much eigenvalue $\lambda_i$ deviates from the statistical properties of its neighboring eigenvalues. Specifically, it computes the deviation of the discrete derivative $d_i = \lambda_{i+1} - \lambda_i$ from the mean of derivatives within windows before and after position $i$, normalized by their respective standard deviations. This normalization produces a significance score $s_i$ that measures how abnormal each eigenvalue gap is relative to its local context. Positions with the highest scores represent points where the spectrum exhibits sudden changes, effectively identifying natural boundaries between different structural components of the graph. The algorithm also handles eigenvalue multiplicity by assigning zero significance to positions where consecutive eigenvalues are identical, ensuring that clusters of repeated eigenvalues remain intact. The identified significant points serve as adaptive partitioning boundaries for the spectrum, allowing PieCoN to focus computational resources on the most informative regions of the eigenvalue distribution. This partitioning preserves sign and basis invariance (as proven in Section 5.2), making our approach robust to different eigenvector calculation methods. A parameter sensitivity analysis for Alg. 1 is presented in Appendix B.

### 4.2 Construction of Constant Filters

For each interval $[a_k, a_{k+1})$, we construct a spectral filter $\boldsymbol{T}_k$, as shown in Fig. 4. The filter is defined as:

$$\boldsymbol{T}_k = \boldsymbol{U}\boldsymbol{E}_k\boldsymbol{U}^\top = \boldsymbol{U}_{[:,a_k:a_{k+1}]}\boldsymbol{U}_{[:,a_k:a_{k+1}]}^\top, \tag{6}$$

where $\boldsymbol{E}_k$ is a binary diagonal matrix with non-zero entries corresponding to the eigenvalues within the $k$-th interval and $\boldsymbol{U}_{[:,a_k:a_{k+1}]}$ is the submatrix of $\boldsymbol{U}$ with columns $a_k$ to $a_{k+1} - 1$. Specifically, the matrix $\boldsymbol{E}_k$ for

---

**Algorithm 1** Thresholding Algorithm for Identifying Significant Eigenvalue Gaps

---
1: **function** IDENTIFY_SIGNIFICANT_GAPS($\boldsymbol{d}$, $\boldsymbol{\lambda}$, $w$, $K$) ▷ $\boldsymbol{d}$: Discrete derivative of eigenvalues, $\boldsymbol{\lambda}$: Sorted eigenvalues, $w$: Window size for averaging, $K$: Number of top indices (Number of spectral intervals)
2:      $\epsilon \leftarrow$ small constant      ▷ A very small positive value to avoid division by zero
3:      $\boldsymbol{s} \leftarrow \boldsymbol{0}$      ▷ Significance of each index
4:      **for** $i \leftarrow w$ to $n - w - 1$ **do**
5:          $\mu_p, \sigma_p \leftarrow \text{mean}(\boldsymbol{d}_{i-w:i}), \text{std}(\boldsymbol{d}_{i-w:i})$      ▷ Mean and standard deviation before $i$
6:          $\mu_n, \sigma_n \leftarrow \text{mean}(\boldsymbol{d}_{i+1:i+w+1}), \text{std}(\boldsymbol{d}_{i+1:i+w+1})$      ▷ Mean and standard deviation after $i$
7:          $s_i \leftarrow \frac{|d - \mu_p|}{\sigma_p + \epsilon} + \frac{|d_i - \mu_n|}{\sigma_n + \epsilon}$      ▷ Sum of normalized distances to adjacent means
8:          **if** $\lambda_i = \lambda_{i-1}$ **then**
9:              $s_i \leftarrow 0$      ▷ Set to zero if no gap exists
10:          **end if**
11:      **end for**
12:      $a_0 = 0$, $a_{K-1} = n + 1$
13:      $a_1, a_2, \ldots, a_{K-2} \leftarrow$ indices of the largest $K - 2$ values in $\boldsymbol{s}$
14:      **return** $a_0, a_1, \ldots, a_{K-1}$
15: **end function**

---

an interval $[a_k, a_{k+1})$ is defined as:

$$\boldsymbol{E}_k = \text{diag}(0, \ldots, \underbrace{1}_{a_k}, 1, \ldots, \underbrace{1}_{a_{k+1}-1}, \ldots, 0). \tag{7}$$

This construction ensures that $\boldsymbol{E}_k$ captures the eigenvectors in the specified interval, allowing the filter $\boldsymbol{T}_k$ to encapsulate the corresponding spectral properties. Our approach differs from traditional polynomial-based spectral filters by enabling more flexible and tailored filtering of the graph's spectral components.

### 4.3 Polynomial Filters

In addition to the spectral filters $\boldsymbol{T}_k$, polynomial filters of the form $\hat{\boldsymbol{A}}^p$ are used (Fig. 5). These filters provide a way to incorporate local neighborhood information into the model. By adjusting the polynomial degree $p$, we can capture varying scales of locality in the graph.

Using the distinct eigenvalues of $\hat{\boldsymbol{A}}$ we can get:

$$(\hat{\boldsymbol{A}} - \lambda_1' \boldsymbol{I}) \cdots (\hat{\boldsymbol{A}} - \lambda_s' \boldsymbol{I}) = 0. \tag{8}$$

This implies that polynomial filters have at most $s$ free parameters because any polynomial of degree higher than $s$ can be reduced to a polynomial of degree $s$. The inclusion of polynomial filters complements the constant spectral filters by providing a smooth interpolation between different spectral components. This combination allows PieCoN to capture both sharp and gradual changes in the graph's spectral properties.

### 4.4 Combining Constant and Polynomial Filters

The final embedding matrix of the nodes is computed by combining the constant spectral filters and polynomial filters. As in Jacobi convolutions (Wang & Zhang, 2022), we apply independent filtering on each of the channels in $\boldsymbol{X}$ simultaneously:

$$\boldsymbol{Z}_{:l} = \underbrace{\sum_{k=0}^{K-1} \alpha_{kl} \boldsymbol{T}_k \sigma(\boldsymbol{X}\boldsymbol{W}^1)\boldsymbol{W}_{:l}^2}_{\text{Constant Filters}} + \underbrace{\sum_{p=0}^{P} \beta_{pl} \hat{\boldsymbol{A}}^p \sigma(\boldsymbol{X}\boldsymbol{W}^1)\boldsymbol{W}_{:l}^2}_{\text{Polynomial Filters}}, \tag{9}$$

where:

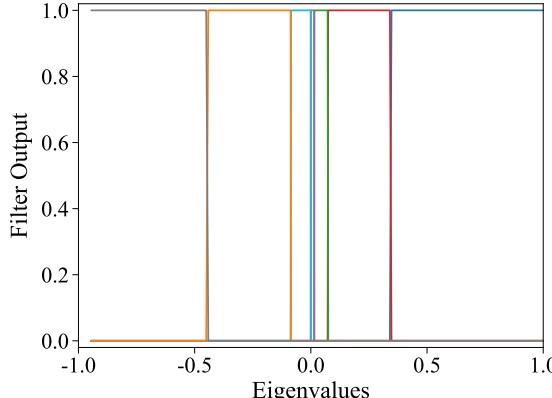

Figure 4: Constant filters.

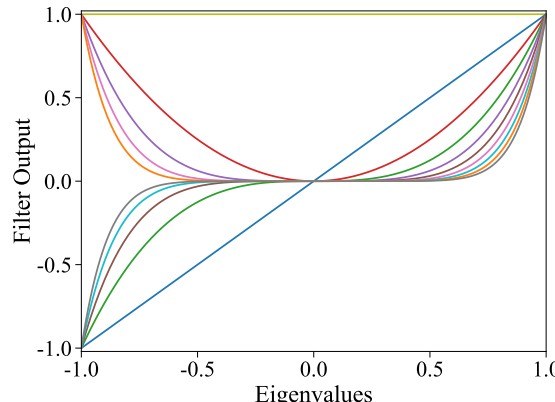

Figure 5: Polynomial filters.

- **Matrix**$_{:l}$ is the $l^{th}$ column of the **Matrix**;
- $\boldsymbol{X}$ is the input feature matrix, representing the initial features of the nodes;
- $\boldsymbol{W}^1$ and $\boldsymbol{W}^2$ are weight matrices to be learned during training. $\boldsymbol{W}^1$ maps the input features to an intermediate space, and $\boldsymbol{W}^2$ maps the intermediate representations to the embedding space;
- $\sigma(\cdot)$ is a non-linear activation function applied element-wise, introducing non-linearity into the model;
- $\alpha_{kl}$ are learnable coefficients associated with the spectral filters $\boldsymbol{T}_k$ for each dimension $l$;
- $\beta_{pl}$ are learnable coefficients associated with the polynomial filters $\hat{\boldsymbol{A}}^p$ for each dimension $l$.

Each element of $\boldsymbol{T}_k$ represents a similarity between two nodes in some range of frequencies. When we perform the matrix multiplication $\boldsymbol{MX}$ for some similarity matrix $\boldsymbol{M} = \boldsymbol{T}_k$, each entry $\boldsymbol{M}_{ij}$ in the similarity matrix $\boldsymbol{M}$ represents the weight or importance of node $j$ in contributing to the feature vector of node $i$. If $\boldsymbol{M}$ is non-negative, it means each node contributes either positively or not at all to the feature aggregation. Negative values, on the other hand, would imply subtracting features from neighbors, which is typically not meaningful in most graph-based learning contexts, where the goal is to aggregate features to enhance node representations. Therefore, we separate $\boldsymbol{T}_k$ into positive and negative parts, as follows:

$$\boldsymbol{Z}_{:l} = \underbrace{\sum_{k=0}^{K-1} \alpha_{kl}^+ (\boldsymbol{T}_k)^+ \sigma(\boldsymbol{XW}^1)\boldsymbol{W}_{:l}^2}_{\text{Positive Part}} + \underbrace{\sum_{k=0}^{K-1} \alpha_{kl}^- (\boldsymbol{T}_k)^- \sigma(\boldsymbol{XW}^1)\boldsymbol{W}_{:l}^2}_{\text{Negative Part}} + \underbrace{\sum_{p=0}^{P} \beta_{pl} \hat{\boldsymbol{A}}^p \sigma(\boldsymbol{XW}^1)\boldsymbol{W}_{:l}^2}_{\text{Polynomial Filters}}, \quad (10)$$

where $\alpha_{kl}^+$ and $\alpha_{kl}^-$ are learnable coefficients associated with the spectral filters $\boldsymbol{T}_k^+$ and $\boldsymbol{T}_k^-$ for each dimension $l$. The superscripts + and - indicate coefficients for the positive and negative parts of the spectral filters, respectively.

It is important to note that while our method could technically be classified as a piecewise polynomial approach, we do not construct separate polynomial functions for each interval. Instead, our approach combines constant functions within specific spectral intervals with global polynomial filters.

## 4.5 Computational Complexity

The computational complexity of our method is broken down as follows:

1. *Eigendecomposition (precomputation):* $\mathcal{O}(n^3)$ for computing spectral components.
2. *Filter construction and sparsification (precomputation):* $\mathcal{O}(n^3 + 2Kn^2 \log(m)) = \mathcal{O}(n^3)$ for constructing and sparsifying $T_K$ filters, where $K$ is the number of spectral intervals, and $m$ is the edge count.

3. *Model propagation:* $\mathcal{O}((K + P)md)$ during training and inference, where $P$ is the polynomial degree, and $d$ is the feature dimension. This matches the theoretical time complexity of JacobiConv (Wang & Zhang, 2022) and GPRGNN (Chien et al., 2021), while being more efficient than BernNet's (He et al., 2021) $\mathcal{O}(P^2md)$. An empirical comparison of the computational efficiency between our approach and the polynomial filtering method JacobiConv is provided in Appendix C.

It is worth noting that while spectral methods like ChebNet (Defferrard et al., 2016), BernNet (He et al., 2021), and JacobiConv (Wang & Zhang, 2022) avoid explicit eigendecomposition, they still face challenges when using higher-degree polynomials (needed for better approximation), as the recursive computation of $\hat{\boldsymbol{A}}^k \boldsymbol{x}$ becomes computationally intensive in practice.

## 5 Theoretical Analysis

This section provides theoretical and empirical analyses to establish the advantages of piecewise constant spectral filters over polynomial filters and validate the design choices of PieCoN. For the theoretical part, we explain an error bound from polynomial approximation theory, which shows the limits of polynomial spectral filtering when dealing with sharp changes in functions. We discuss the challenges with eigenvector representations, such as their invariance under sign flips and basis shifts, which can affect the generalization of graph learning models. Additionally, in Appendix D, we analyze how specific graph structures influence the eigenvalue spectrum, with a focus on the eigenvalue 0, and explain why separating constant filters into negative and positive parts helps improve model performance and reduces approximation errors.

### 5.1 Error Analysis for Polynomial Approximation

To analyze the fundamental limitations of polynomial approximations in spectral filtering, we establish a theorem characterizing the error bounds.

**Theorem 1** (Approximation error for $\epsilon$-dense eigenvalues). *Let $\hat{\boldsymbol{A}} \in \mathbb{R}^{n \times n}$ be a normalized adjacency matrix with spectrum $\{\lambda_i\}_{i=1}^n$ where $-1 \leq \lambda_1 \leq \cdots \leq \lambda_n \leq 1$. Assume that these eigenvalues are $\epsilon$-dense on $[-1, 1]$ and that $d^2\epsilon < 1$. Let $f : [-1, 1] \to \mathbb{R}$ be a filter function with $\|f\|_\infty = \sup_{x \in [-1,1]} |f(x)| = 1$. For any polynomial $p \in \mathcal{P}_d$(the space of polynomials of degree at most d), the approximation error is:*

$$\mathcal{E}(p, f) = \sum_{i=1}^n |p(\lambda_i) - f(\lambda_i)| \geq \|p\|_\infty (1 - d^2\epsilon) - 1. \tag{11}$$

Generally, as $n$ increases, the eigenvalues become more densely packed in $[-1, 1]$, causing $\epsilon$ to approach zero. This means $d^2\epsilon$ will also approach zero, making the lower bound on the approximation error converge to $\|p\|_\infty - 1$. Therefore, to minimize the error bound while maintaining sufficient approximation power, setting $\|p\|_\infty \leq 1$ is a natural choice since the target function satisfies $\|f\|_\infty = 1$. This normalization allows for fair comparison between different polynomial approximations and simplifies the analysis. Hence, in the following theorem, we specifically focus on polynomials with $\|p\|_\infty \leq 1$ to analyze the particular challenge of approximating functions with jump discontinuities.

**Theorem 2** (Approximation error for functions with jump discontinuities). *Let $\hat{\boldsymbol{A}} \in \mathbb{R}^{n \times n}$ be a normalized adjacency matrix with spectrum $\{\lambda_i\}_{i=1}^n$ where $-1 \leq \lambda_1 \leq \cdots \leq \lambda_n \leq 1$. Let $f : [-1, 1] \to \mathbb{R}$ be some filter function with $\|f\|_\infty = 1$ that the model needs to find and suppose it has a jump discontinuity of magnitude $h > 0$ between consecutive eigenvalues $\lambda_R$ and $\lambda_{R+1}$ ($|f(\lambda_{R+1}) - f(\lambda_R)| = h$). For any polynomial $p \in \mathcal{P}_d$(the space of polynomials of degree at most d), satisfying $\|p\|_\infty \leq 1$, the approximation error is:*

$$\mathcal{E}(p, f) = \sum_{i=1}^n |p(\lambda_i) - f(\lambda_i)| \geq h - |\lambda_{R+1} - \lambda_R| \cdot d^2. \tag{12}$$

This result demonstrates a key limitation: For small $d$ when $|\lambda_{R+1} - \lambda_R| \ll \frac{h}{d^2}$, the approximation error $\mathcal{E}(p, f) \geq \mathcal{E}_2 \approx h$. This is particularly problematic for spectral filtering where: (1) sharp transitions in the

Table 1: Statistics of the datasets used for node classification. $\nu(0)$ is the multiplicity of eigenvalue 0.

| Dataset | Nodes | Edges | Classes | Homophily Ratio | $\nu(0)$/Nodes |
|---|---|---|---|---|---|
| Chameleon | 2,277 | 31,396 | 5 | 0.23 | 0.52 |
| Squirrel | 5,201 | 198,423 | 5 | 0.22 | 0.37 |
| Actor | 7,600 | 26,705 | 5 | 0.22 | 0.15 |
| Amazon-Ratings | 24,492 | 93,050 | 5 | 0.38 | 0.17 |
| Texas | 183 | 168 | 5 | 0.11 | 0.35 |
| Cora | 2,708 | 5,278 | 7 | 0.81 | 0.11 |
| Citeseer | 3,327 | 4,614 | 6 | 0.74 | 0.14 |
| Amazon-Photo | 7,650 | 71,831 | 8 | 0.83 | 0.02 |
| Pubmed | 19,717 | 44,324 | 3 | 0.80 | 0.61 |

filter response are often desired ($h$ is large), (2) some eigenvalues may be very close together ($|\lambda_{R+1} - \lambda_R|$ is small), and (3) using high-degree polynomials is computationally expensive.

In contrast, by using piecewise constant filters, we can add a constant filter only at point $\lambda_{R+1}$ with the value $h$ and eliminate the jump entirely. The ablation study in Table 3 demonstrates that adding constant filters to polynomial filters improves performance. This theoretical result justifies combining polynomial filters with constant filters in our approach.

### 5.2 Sign and Basis Invariance

Eigenvectors corresponding to a given eigenvalue can have multiple representations. For example, if $\lambda = 0$ is an eigenvalue of the normalized adjacency matrix $\hat{\boldsymbol{A}}$, any eigenvector $\boldsymbol{u}_i$ associated with $\lambda = 0$ can be replaced with its opposite $-\boldsymbol{u}_i$ or any linear combination of eigenvectors for the same eigenvalue. This variability introduces sign and basis ambiguity problems, which can lead to inconsistent or unpredictable results in learning tasks.

**Proposition 1** (This follows directly from (Lim et al., 2023)). *Polynomial filters are invariant to sign changes and basis choices.*

The invariance of polynomial filters stems from the stability of eigenvector products $\boldsymbol{U}_\mu \boldsymbol{U}_\mu^\top$ under different basis representations. This property ensures that matrix powers $\hat{\boldsymbol{A}}^p$ maintain consistent behavior regardless of the specific eigenbasis chosen, making polynomial filters reliable for spectral graph operations.

**Proposition 2.** *Constant filters are invariant to sign changes and basis choices.*

Both polynomial and constant filters are robust to different eigenvector representations, ensuring that learned representations are not affected by arbitrary sign changes or basis choices, thus improving model stability and generalization. By leveraging filters with this characteristic, our model can produce consistent outputs despite the inherent ambiguities in eigendecomposition.

## 6 Experimental Evaluation

### 6.1 Datasets

We evaluate PieCoN on seven diverse node classification datasets with varying graph structures and homophily ratios (Table 1). Cora, Citeseer, and Pubmed are citation networks where nodes are research papers and edges represent citations. Photo is a product co-occurrence graph with nodes as products and edges representing co-purchase relationships. Actor is a graph where nodes are actors and edges denote co-occurrence in films. Chameleon and Squirrel are graphs derived from Wikipedia pages. Nodes represent web pages, and edges denote mutual links. Texas is an academic web graph where nodes are webpages from the University of Texas and edges represent hyperlinks between pages. Amazon-Ratings is a product co-purchasing network

Table 2: Results on real-world node classification tasks.

| Model | Heterophilic | | | | | Homophilic | | | |
|---|---|---|---|---|---|---|---|---|---|
| | Chameleon | Squirrel | Actor | Amazon-Ratings | Texas | Cora | Citeseer | Amazon-Photo | Pubmed |
| *Spatial-based GNNs* | | | | | | | | | |
| GCN | $68.10^{***}_{\pm1.20}$ | $50.11^{***}_{\pm1.21}$ | $34.65^{***}_{\pm0.68}$ | $48.80^{***}_{\pm0.22}$ | $78.69^{***}_{\pm1.80}$ | $87.18^{***}_{\pm0.87}$ | $81.04_{\pm0.67}$ | $85.87^{***}_{\pm0.83}$ | $87.31^{***}_{\pm0.31}$ |
| GAT | $63.13^{***}_{\pm1.93}$ | $44.49^{***}_{\pm0.88}$ | $33.93^{***}_{\pm2.47}$ | $50.28^{***}_{\pm0.55}$ | $77.54^{***}_{\pm0.98}$ | $88.03^{***}_{\pm0.79}$ | $80.52_{\pm0.71}$ | $90.94^{***}_{\pm0.68}$ | $87.29^{***}_{\pm0.48}$ |
| H$_2$GCN | $57.11^{***}_{\pm1.58}$ | $36.42^{***}_{\pm1.89}$ | $35.86^{***}_{\pm1.03}$ | $48.17^{***}_{\pm0.52}$ | $88.36_{\pm2.62}$ | $86.92^{***}_{\pm1.37}$ | $77.07^{***}_{\pm1.64}$ | $93.02^{***}_{\pm0.91}$ | $88.93^{***}_{\pm0.35}$ |
| GCNII | $63.44^{***}_{\pm0.85}$ | $41.96^{***}_{\pm1.02}$ | $36.89^{***}_{\pm0.95}$ | $46.60^{***}_{\pm1.20}$ | $89.18_{\pm4.43}$ | $88.46_{\pm0.82}$ | $79.97^{***}_{\pm0.65}$ | $89.94^{***}_{\pm0.31}$ | $89.68^{***}_{\pm0.30}$ |
| *Spectral-based GNNs* | | | | | | | | | |
| Free eigenvalues | $69.58^{***}_{\pm1.31}$ | $59.76^{***}_{\pm1.01}$ | $41.61^{***}_{\pm0.63}$ | $44.28^{***}_{\pm1.13}$ | $88.20_{\pm3.28}$ | $84.91^{***}_{\pm0.89}$ | $77.39^{***}_{\pm0.82}$ | $86.08^{***}_{\pm0.81}$ | $86.07^{***}_{\pm0.47}$ |
| LanczosNet | $64.81^{***}_{\pm1.56}$ | $48.64^{***}_{\pm1.77}$ | $38.16^{*}_{\pm0.91}$ | $48.35^{***}_{\pm0.40}$ | $76.39^{***}_{\pm4.43}$ | $87.77_{\pm1.45}$ | $80.05_{\pm1.65}$ | $93.21^{***}_{\pm0.85}$ | $84.41^{***}_{\pm0.66}$ |
| ChebyNet | $59.28^{***}_{\pm1.25}$ | $40.55^{***}_{\pm0.42}$ | $37.61^{***}_{\pm0.89}$ | $50.20^{***}_{\pm0.52}$ | $77.21^{***}_{\pm2.95}$ | $86.67^{***}_{\pm0.82}$ | $79.11^{***}_{\pm0.75}$ | $93.77^{***}_{\pm0.32}$ | $90.11^{***}_{\pm0.26}$ |
| GPR-GNN | $67.28^{***}_{\pm1.09}$ | $50.15^{***}_{\pm1.92}$ | $39.92_{\pm0.67}$ | $49.37^{***}_{\pm0.71}$ | $88.53_{\pm3.36}$ | $88.57_{\pm0.69}$ | $80.12_{\pm0.83}$ | $93.85^{***}_{\pm0.28}$ | $91.36_{\pm0.40}$ |
| BernNet | $68.29^{***}_{\pm1.58}$ | $51.35^{***}_{\pm0.73}$ | $41.79^{***}_{\pm1.01}$ | $48.82^{***}_{\pm0.20}$ | $89.02_{\pm3.45}$ | $88.52_{\pm0.95}$ | $80.09_{\pm0.79}$ | $93.63^{***}_{\pm0.35}$ | $88.98_{\pm0.46}$ |
| PPGNN | $69.45^{***}_{\pm1.05}$ | $48.47^{***}_{\pm2.51}$ | $39.65_{\pm0.66}$ | $47.96^{***}_{\pm0.89}$ | $85.57_{\pm2.62}$ | $88.32_{\pm0.60}$ | $80.98_{\pm0.51}$ | $95.09^{*}_{\pm0.31}$ | $90.11_{\pm0.26}$ |
| ChebNetII | $71.37^{***}_{\pm1.01}$ | $57.72^{***}_{\pm0.59}$ | $41.75^{*}_{\pm1.07}$ | $48.79^{***}_{\pm0.21}$ | $89.11_{\pm3.43}$ | $88.71_{\pm0.93}$ | $80.53_{\pm0.79}$ | $94.92^{*}_{\pm0.33}$ | $89.76^{*}_{\pm0.32}$ |
| JacobiConv | $73.92^{*}_{\pm1.07}$ | $57.38^{***}_{\pm0.60}$ | $40.43_{\pm0.81}$ | $48.53^{***}_{\pm0.96}$ | $89.02_{\pm2.79}$ | $88.69_{\pm1.03}$ | $81.65^{*}_{\pm0.46}$ | $95.36_{\pm0.24}$ | $87.83^{***}_{\pm0.43}$ |
| DSF-Jacobi-R | $72.17^{***}_{\pm0.79}$ | $55.84^{***}_{\pm0.94}$ | $39.89_{\pm0.54}$ | $48.68^{***}_{\pm0.34}$ | $89.18_{\pm3.44}$ | $88.31_{\pm0.89}$ | $81.11_{\pm0.63}$ | $94.90^{*}_{\pm0.31}$ | $88.97^{***}_{\pm0.39}$ |
| OptBasisGNN | $74.40^{*}_{\pm0.90}$ | $63.98^{*}_{\pm1.12}$ | $\mathbf{42.39^{***}_{\pm0.52}}$ | $48.80^{***}_{\pm0.21}$ | $87.38_{\pm2.79}$ | $87.96^{*}_{\pm0.71}$ | $80.79_{\pm1.35}$ | $94.71^{***}_{\pm0.33}$ | $87.36^{***}_{\pm0.41}$ |
| Specformer | $74.92_{\pm0.98}$ | $64.26_{\pm1.18}$ | $41.56_{\pm1.25}$ | OOM | $83.60^{*}_{\pm3.27}$ | $87.55^{*}_{\pm0.87}$ | $80.98_{\pm0.79}$ | $95.29_{\pm0.30}$ | OOM |
| UniFilter | $74.11_{\pm1.68}$ | $63.52^{*}_{\pm1.30}$ | $40.11_{\pm1.31}$ | $50.02^{***}_{\pm0.70}$ | $86.72_{\pm3.77}$ | $89.10_{\pm1.07}$ | $81.21_{\pm1.66}$ | $94.96_{\pm0.74}$ | $91.36_{\pm0.45}$ |
| PieCoN | $\mathbf{75.75_{\pm0.96}}$ | $\mathbf{65.67_{\pm0.82}}$ | $39.79_{\pm0.56}$ | $\mathbf{52.37_{\pm0.50}}$ | $\mathbf{89.34_{\pm3.11}}$ | $\mathbf{89.16_{\pm0.64}}$ | $80.98_{\pm0.57}$ | $\mathbf{95.65_{\pm0.34}}$ | $\mathbf{91.39_{\pm0.41}}$ |

$^{*}p < 0.05$ (significant difference from PieCoN)
$^{***}p < 0.001$ (highly significant difference from PieCoN)

where nodes are products and edges indicate frequent co-purchases, with the task of predicting product rating classes. The ratio of eigenvalue 0 multiplicity to the number of nodes is shown in the last column of Table 1.

All datasets were randomly split into 60% training, 20% validation, and 20% test sets for 10 different seeds. For each dataset, we report the average performance along with the 95% confidence interval. Details about hyperparameter optimization and the running environment are provided in Appendix E.

## 6.2 Baseline Models

We compare PieCoN against several baseline models categorized into different groups based on their underlying graph learning methodologies:

- **Spatial-based GNNs**: Graph Convolutional Network (GCN) (Kipf & Welling, 2017), Graph Attention Network (GAT) (Veličković et al., 2018), Higher-order GCN (H$_2$GCN) (Zhu et al., 2020), and GCNII (Chen et al., 2020).

- **Spectral-based GNNs**: UniFilter (Huang et al., 2024), LanczosNet (Liao et al., 2019), ChebyNet (Defferrard et al., 2016), Generalized PageRank GNN (GPR-GNN) (Chien et al., 2021), BernNet (He et al., 2021), PP-GNN (Lingam et al., 2022), ChebNetII (He et al., 2022), DSF-Jacobi-R (Guo et al., 2023), JacobiConv (Wang & Zhang, 2022), OptBasisGNN (Guo & Wei, 2023), and Specformer (Bo et al., 2023).

- **Free eigenvalues**: A graph neural network that learns a spectral filter by directly parameterizing the eigenspectrum $\hat{A} = U\Lambda U^{\top}$, where $\Lambda$ contains trainable eigenvalues.

## 6.3 Results

Table 2 shows the node classification accuracy of PieCoN compared to baseline models across various datasets. We observe that PieCoN achieves the highest performance on seven datasets. Notably, the largest

Table 3: Ablation study results.

| Pos. Part | Neg. Part | Poly. | Chameleon | Squirrel | Actor | Amazon-Ratings | Texas | Cora | Citeseer | Amazon-Photo | Pubmed |
|---|---|---|---|---|---|---|---|---|---|---|---|
| ✗ | ✗ | ✓ | $66.35^{***}_{\pm0.88}$ | $48.39^{***}_{\pm0.78}$ | $\mathbf{40.31}_{\pm0.94}$ | $50.21^{***}_{\pm0.87}$ | $88.69_{\pm3.28}$ | $88.74_{\pm0.77}$ | $80.33_{\pm0.85}$ | $95.57_{\pm0.35}$ | $91.24_{\pm0.40}$ |
| ✗ | ✓ | ✗ | $67.26^{***}_{\pm0.50}$ | $54.65^{***}_{\pm0.72}$ | $32.07^{***}_{\pm1.17}$ | $49.66^{***}_{\pm0.71}$ | $\mathbf{90.66}_{\pm2.30}$ | $84.30^{***}_{\pm1.06}$ | $75.45^{***}_{\pm0.68}$ | $92.70^{***}_{\pm0.34}$ | $90.72^{*}_{\pm0.39}$ |
| ✓ | ✗ | ✗ | $73.61^{***}_{\pm0.81}$ | $60.50^{***}_{\pm1.03}$ | $38.98_{\pm0.78}$ | $47.35^{***}_{\pm0.76}$ | $90.00_{\pm3.11}$ | $86.80^{***}_{\pm1.16}$ | $\mathbf{81.72}_{\pm0.58}$ | $94.95^{*}_{\pm0.33}$ | $90.74^{*}_{\pm0.48}$ |
| ✓ | ✓ | ✗ | $74.77_{\pm1.01}$ | $65.00_{\pm1.12}$ | $39.02_{\pm0.54}$ | $49.28^{***}_{\pm0.62}$ | $89.67_{\pm2.46}$ | $87.22^{*}_{\pm1.12}$ | $81.54_{\pm0.63}$ | $94.76^{***}_{\pm0.39}$ | $90.83_{\pm0.41}$ |
| ✓ | ✓ | ✓ | $\mathbf{75.75}_{\pm0.96}$ | $\mathbf{65.67}_{\pm0.82}$ | $39.79_{\pm0.56}$ | $\mathbf{52.37}_{\pm0.50}$ | $89.34_{\pm3.11}$ | $\mathbf{89.16}_{\pm0.64}$ | $80.98_{\pm0.57}$ | $\mathbf{95.65}_{\pm0.34}$ | $\mathbf{91.38}_{\pm0.41}$ |

$^{*}p < 0.05$ (significant difference from full model)
$^{***}p < 0.001$ (highly significant difference from full model)

improvements are observed on the heterophilic datasets Chameleon, Squirrel, and Amazon-Ratings, with gains of 0.83%, 1.41%, and 2.09%, respectively. This may be linked to the high multiplicity of the eigenvalue 0 in the normalized adjacency matrix of these graphs (see Table 1), to which our method gives more importance.

For homophilic datasets, such as Cora, Amazon-Photo, and Pubmed, PieCoN also demonstrates competitive performance, achieving slight improvements over existing methods. The smaller gains suggest that traditional GNNs already perform well in these settings, as they inherently align with homophilic assumptions. Nonetheless, PieCoN remains robust, indicating that its spectral filtering approach does not hinder its ability to learn from homophilic graphs. These results indicate that PieCoN is effective in both heterophilic and homophilic settings.

We also run $t$-tests comparing PieCoN with each baseline method to verify the statistical significance of our results. Stars in Table 2 show when a baseline performs significantly different than PieCoN (* for $p < 0.05$, *** for $p < 0.001$). Most baselines show significant differences on heterophilic datasets, confirming that our gains are meaningful. For example, all methods except Specformer and UniFilter show significantly lower performance on Chameleon. On homophilic datasets, several methods show no significant difference from PieCoN, indicating competitive but not always superior performance.

## 6.4 Ablation Study

We have performed an ablation study using Eq. (10) to evaluate the contribution of each component on model performance. The results in Table 3 reveal several key findings. First, the full model incorporating all three components (positive part, negative part, and polynomial filters) achieves the best performance on 6 out of 9 datasets, with notable improvements on Chameleon (75.75%), Squirrel (65.67%), and Amazon-Ratings (52.37%). The combination of positive and negative parts without polynomial filters also shows strong performance, suggesting that these components capture complementary spectral information. For instance, on Squirrel, adding the negative part to the positive part improves accuracy from 60.50% to 65.00%. Interestingly, on the Actor dataset, using only polynomial filters yields the best performance (40.31%), while on Citeseer, the positive part alone achieves optimal results (81.72%).

In a separate experiment, we also create a simple spectral method with the eigenvalues as parameters. The results of this experiment are presented in Table 2 with the model name "Free eigenvalues". However, this approach may be less effective because the method does not receive any explicit structural information associated with the eigenvalues.

## 7 Limitations

Our work has some limitations. The computational complexity of $\mathcal{O}(n^3)$ for eigendecomposition presents scalability challenges for large-scale graphs. However, this preprocessing step occurs only once and has become increasingly practical with modern hardware, taking approximately 8 minutes for datasets with 25,000 nodes on an NVIDIA A100 GPU. It is worth noting that our approach is designed for graphs of reasonable size (up to tens of thousands of nodes), where the eigendecomposition is a worthwhile one-time preprocessing cost

given the expressivity benefits. Also, our work does not address the problem of scaling GNNs to massive graphs with millions of nodes, which is a separate research area requiring specialized techniques like graph sampling or partitioning beyond the scope of this paper. Furthermore, the model's performance is highly dependent on how we partition the eigenvalue intervals, and our current approach using hard thresholding to identify significant spectral changes may not be optimal.

## 8 Conclusion

In this paper, we presented the Piecewise Constant Spectral Graph Neural Network (PieCoN), a new approach to graph prediction tasks. Our method aims to address some limitations of existing spectral GNNs by combining constant spectral filters with polynomial filters to capture a broader range of spectral characteristics in real-world graphs. We introduced an adaptive spectral partitioning technique that analyzes the derivative of sorted eigenvalues to identify significant spectral changes. This helps focus on the most informative regions of the spectrum. PieCoN expands the search space of possible eigenvalue filters beyond traditional polynomial-based filters, allowing for a more tailored capture of graph spectral properties. This is particularly useful when dealing with graphs that have high eigenvalue multiplicity. By integrating spectral filters with polynomial filters, our approach attempts to model both global graph structure and local neighborhood information. Our experiments on nine benchmark datasets, covering both homophilic and heterophilic graph structures, suggest that PieCoN performs well on both types of datasets.

## Acknowledgements

Vahan Martirosyan is the recipient of a PhD scholarship from the STIC Doctoral School of Université Paris-Saclay.

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

# A   Proofs

**Theorem 1** (Approximation error for $\epsilon$-dense eigenvalues)**.** *Let $\hat{\boldsymbol{A}} \in \mathbb{R}^{n \times n}$ be a normalized adjacency matrix with spectrum $\{\lambda_i\}_{i=1}^n$ where $-1 \leq \lambda_1 \leq \cdots \leq \lambda_n \leq 1$. Assume that these eigenvalues are $\epsilon$-dense on $[-1,1]$ and that $d^2\epsilon < 1$. Let $f : [-1,1] \to \mathbb{R}$ be a filter function with $\|f\|_\infty = \sup_{x \in [-1,1]} |f(x)| = 1$. For any polynomial $p \in \mathcal{P}_d$ (the space of polynomials of degree at most $d$) the approximation error is:*

$$\mathcal{E}(p,f) = \sum_{i=1}^n |p(\lambda_i) - f(\lambda_i)| \geq \|p\|_\infty (1 - d^2\epsilon) - 1. \tag{13}$$

*Proof.* Let $x_0 \in [-1,1]$ be a point where $|p(x_0)| = \|p\|_\infty$, *i.e.*, the point where the polynomial $p$ attains its maximum absolute value on the interval $[-1,1]$. Since the eigenvalues $\{\lambda_i\}_{i=1}^n$ are $\epsilon$-dense on $[-1,1]$, there exists an eigenvalue $\lambda_j$ such that $|x_0 - \lambda_j| \leq \epsilon$. By Markov's polynomial inequality (Sahoo & Riedel, 1998):

$$\|p'\|_\infty \leq d^2 \|p\|_\infty. \tag{14}$$

Using the mean value theorem (Achiezer, 1992), there exists $\xi \in [x_0, \lambda_j]$ (or $[\lambda_j, x_0]$ if $\lambda_j > x_0$) such that:

$$|p(x_0) - p(\lambda_j)| = |p'(\xi)| \cdot |x_0 - \lambda_j| \leq \|p'\|_\infty \cdot |x_0 - \lambda_j| \leq d^2 \|p\|_\infty \cdot \epsilon. \tag{15}$$

Therefore,

$$|p(\lambda_j)| \geq |p(x_0)| - |p(x_0) - p(\lambda_j)| \geq \|p\|_\infty - d^2 \|p\|_\infty \epsilon = \|p\|_\infty (1 - d^2\epsilon). \tag{16}$$

Since $\|f\|_\infty = 1$, we know that $|f(\lambda_j)| \leq 1$. Then,

$$|p(\lambda_j) - f(\lambda_j)| \geq |p(\lambda_j)| - |f(\lambda_j)| \geq \|p\|_\infty (1 - d^2\epsilon) - 1. \tag{17}$$

Since $\mathcal{E}(p,f) = \sum_{i=1}^n |p(\lambda_i) - f(\lambda_i)| \geq |p(\lambda_j) - f(\lambda_j)|$, we have:

$$\mathcal{E}(p,f) \geq \|p\|_\infty (1 - d^2\epsilon) - 1. \tag{18}$$

This establishes the claimed lower bound on the approximation error. $\qquad\square$

**Theorem 2** (Approximation error for functions with jump discontinuities)**.** *Let $\hat{\boldsymbol{A}} \in \mathbb{R}^{n \times n}$ be a normalized adjacency matrix with spectrum $\{\lambda_i\}_{i=1}^n$ where $-1 \leq \lambda_1 \leq \cdots \leq \lambda_n \leq 1$. Let $f : [-1,1] \to \mathbb{R}$ be some filter function with $\|f\|_\infty = 1$ that the model needs to find and suppose it has a jump discontinuity of magnitude $h > 0$ between consecutive eigenvalues $\lambda_R$ and $\lambda_{R+1}$. For any polynomial $p \in \mathcal{P}_d$ (the space of polynomials of degree at most $d$), satisfying $\|p\|_\infty \leq 1$, the approximation error is:*

$$\mathcal{E}(p,f) = \sum_{i=1}^n |p(\lambda_i) - f(\lambda_i)| \geq h - |\lambda_{R+1} - \lambda_R| \cdot d^2. \tag{19}$$

*Proof.* By Markov's polynomial inequality (Sahoo & Riedel, 1998):

$$\|p'\|_\infty \leq d^2 \|p\|_\infty \leq d^2, \quad \forall p \in \mathcal{P}_d. \tag{20}$$

Next using the mean value theorem (Achiezer, 1992), we find that $\exists\, \xi_{xy} \in [x,y]$ such that:

$$|p(x) - p(y)| = |p'(\xi_{xy})| \cdot |x - y| \leq d^2 |x - y|, \quad \forall\, x, y \in [-1,1]. \tag{21}$$

Now, we define the local error at consecutive eigenvalues as:

$$\mathcal{E}_2 = |p(\lambda_R) - f(\lambda_R)| + |p(\lambda_{R+1}) - f(\lambda_{R+1})|. \tag{22}$$

Applying the triangle inequality, we obtain:

$$\begin{aligned} \mathcal{E}_2 &\geq |f(\lambda_{R+1}) - f(\lambda_R)| - |p(\lambda_{R+1}) - p(\lambda_R)| \\ &\geq h - |\lambda_{R+1} - \lambda_R| \cdot d^2. \end{aligned} \tag{23}$$

Since it holds that $\mathcal{E}(p,f) \geq \mathcal{E}_2$, the result follows. $\qquad\square$

**Proposition 1** (This follows directly from (Lim et al., 2023)). *Polynomial filters are invariant to sign changes and basis choices.*

*Proof.* The key property we use is that the product $\boldsymbol{U}_\mu \boldsymbol{U}_\mu^\top$, where $\boldsymbol{U}_\mu$ is a matrix of eigenvectors associated with eigenvalue $\mu$, remains invariant under sign changes and different choices of basis (Lim et al., 2023). This implies that regardless of which orthonormal basis is chosen for a given eigenspace, the product $\boldsymbol{U}_\mu \boldsymbol{U}_\mu^\top$ stays the same.

Consider two orthonormal bases $\boldsymbol{U}_\mu$ and $\boldsymbol{V}_\mu$ for the eigenspace corresponding to eigenvalue $\mu$. There exists an orthogonal matrix $\boldsymbol{Q}$ such that:

$$\boldsymbol{V}_\mu = \boldsymbol{U}_\mu \boldsymbol{Q}. \tag{24}$$

Using this, we show the invariance:

$$\boldsymbol{V}_\mu \boldsymbol{V}_\mu^\top = (\boldsymbol{U}_\mu \boldsymbol{Q})(\boldsymbol{U}_\mu \boldsymbol{Q})^\top = \boldsymbol{U}_\mu \boldsymbol{Q} \boldsymbol{Q}^\top \boldsymbol{U}_\mu^\top = \boldsymbol{U}_\mu \boldsymbol{U}_\mu^\top. \tag{25}$$

where we used the fact that $\boldsymbol{Q}\boldsymbol{Q}^\top = \boldsymbol{I}$ since $\boldsymbol{Q}$ is orthogonal. This confirms that $\boldsymbol{U}_\mu \boldsymbol{U}_\mu^\top$ is invariant under the change of basis.

Now, consider the matrix power $\hat{\boldsymbol{A}}^p$, which can be expressed as:

$$\hat{\boldsymbol{A}}^p = \sum_{i=1}^n \lambda_i^p \boldsymbol{u}_i \boldsymbol{u}_i^\top = \sum_{i=1}^s (\lambda_i')^p \boldsymbol{U}_{\lambda_i'} \boldsymbol{U}_{\lambda_i'}^\top, \tag{26}$$

where $\lambda_i'$ are the distinct eigenvalues, and $\boldsymbol{U}_{\lambda_i'}$ are the corresponding eigenvector matrices.

Since each term $\boldsymbol{U}_{\lambda_i'} \boldsymbol{U}_{\lambda_i'}^\top$ is invariant to basis choices and each term $\boldsymbol{u}_i \boldsymbol{u}_i^\top = (-\boldsymbol{u}_i)(-\boldsymbol{u}_i)^\top$ is invariant to sign changes, it follows that $\hat{\boldsymbol{A}}^p$ is also invariant to both. $\square$

**Proposition 2.** *Constant filters are invariant to sign changes and basis choices.*

*Proof.* Consider the constant filter $\boldsymbol{T}_k$ defined over the interval $[a_k, a_{k+1})$ for some $k$. Let $t_k$ be the index such that $\lambda_{t_k}' = \lambda_{a_k}$, and thus $\lambda_{t_{k+1}-1}' = \lambda_{a_{k+1}-1}$.

Suppose $l, r$ are indices such that $\lambda_{l-1} \neq \lambda_l = \lambda_{l+1} = \cdots = \lambda_{r-1} \neq \lambda_r$. According to Alg. 1, no significant point $i$ will be selected with $l < i < r$, ensuring that constant eigenvalue intervals are not split. Consequently, $\lambda_{t_k}' \neq \lambda_{a_k-1}$ when $a_k > 1$, and $\lambda_{t_{k+1}-1}' \neq \lambda_{a_{k+1}}$ when $a_{k+1} \leq n$.

The distinct eigenvalues of $\hat{\boldsymbol{A}}$ within the interval $[\lambda_{a_k}, \lambda_{a_{k+1}-1}]$ are:

$$\lambda_{t_k}', \lambda_{t_k+1}', \ldots, \lambda_{t_{k+1}-1}'. \tag{27}$$

Using Equation 6, we express $\boldsymbol{T}_k$ as:

$$\boldsymbol{T}_k = \boldsymbol{U}_{[:,a_k:a_{k+1}]} \boldsymbol{U}_{[:,a_k:a_{k+1}]}^\top = \sum_{i=a_k}^{a_{k+1}-1} \boldsymbol{u}_{\lambda_i} \boldsymbol{u}_{\lambda_i}^\top = \sum_{i=t_k}^{t_{k+1}-1} \boldsymbol{U}_{\lambda_i'} \boldsymbol{U}_{\lambda_i'}^\top. \tag{28}$$

Since each term $\boldsymbol{U}_{\lambda_i'} \boldsymbol{U}_{\lambda_i'}^\top$ is invariant to basis choices and each term $\boldsymbol{u}_i \boldsymbol{u}_i^\top = (-\boldsymbol{u}_i)(-\boldsymbol{u}_i)^\top$ is invariant to sign changes, it follows that $\boldsymbol{T}_k$ is also invariant to both. $\square$

## B  Parameter Sensitivity Analysis of Algorithm 1

We further investigate the impact of the window size ($w$) and the number of limits ($K$) from Alg. 1 on model performance. To isolate the effect of each parameter, we conduct controlled experiments where we fix one parameter at its optimal value while varying the other. Specifically, when examining window size effects, we fix $K$ at its optimal value for each dataset, and when studying the number of spectral intervals,

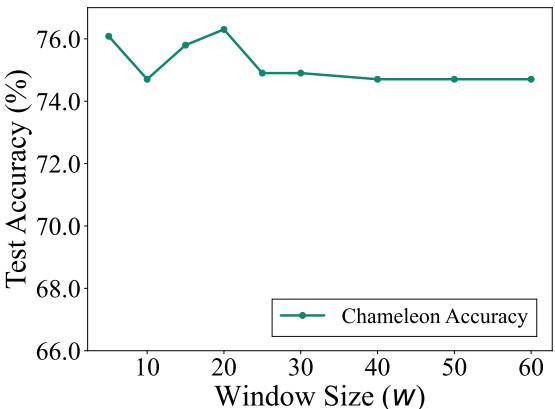

Figure 6: Effect of $w$ on Chameleon dataset performance with optimal $K$.

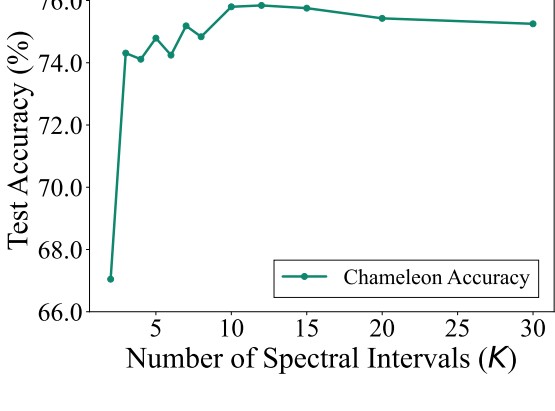

Figure 7: Effect of $K$ on Chameleon dataset performance with optimal $w$.

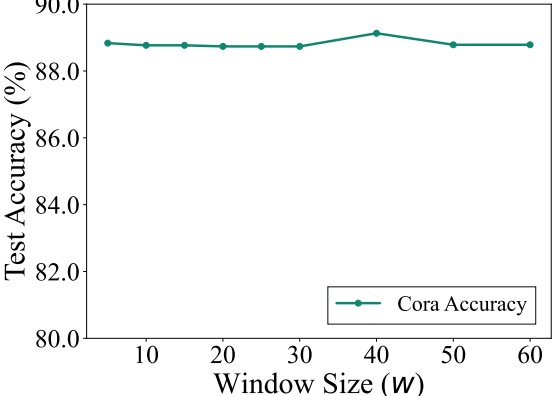

Figure 8: Effect of $w$ on Cora dataset performance with optimal $K$.

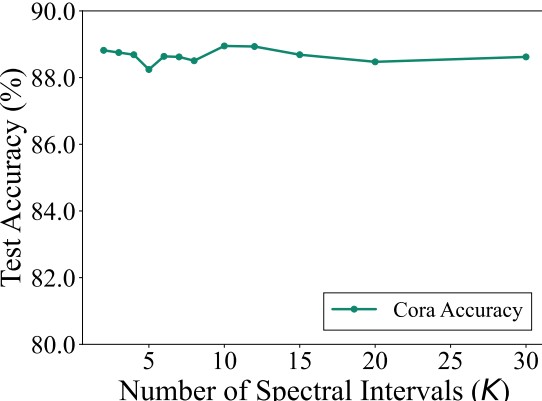

Figure 9: Effect of $K$ on Cora dataset performance with optimal $w$.

we fix $w$ at its optimal value. As shown in Fig. 6 and Fig. 7, the Chameleon dataset exhibits distinctive parameter sensitivities. Performance peaks at $w = 20$ with an accuracy of 76.30%, followed by a sharp decline and stabilization around 74.70% for larger window sizes. For the number of spectral intervals, we observe a dramatic performance increase from $K = 2$ to $K = 10$, with optimal performance in the range $10 \leq K \leq 15$ (reaching 75.84%), followed by a gradual decrease as $K$ increases further. In contrast, Cora (Fig. 8 and Fig. 9) shows more stability across different window sizes with an optimal value at $w = 40$ reaching 89.13% accuracy. For spectral intervals, Cora demonstrates a similar pattern to Chameleon but with less pronounced differences, showing peak performance at $10 \leq K \leq 12$.

These results indicate that the number of spectral intervals ($K$) significantly influences model performance, whereas window size ($w$) has a more limited effect. Too few partitions fail to capture important spectral characteristics, while too many may introduce noise.

## C  Computational Efficiency Comparison

To evaluate the computational efficiency of our approach, we conduct a running time analysis comparing the constant filter component of PieCoN with JacobiConv's polynomial filters.

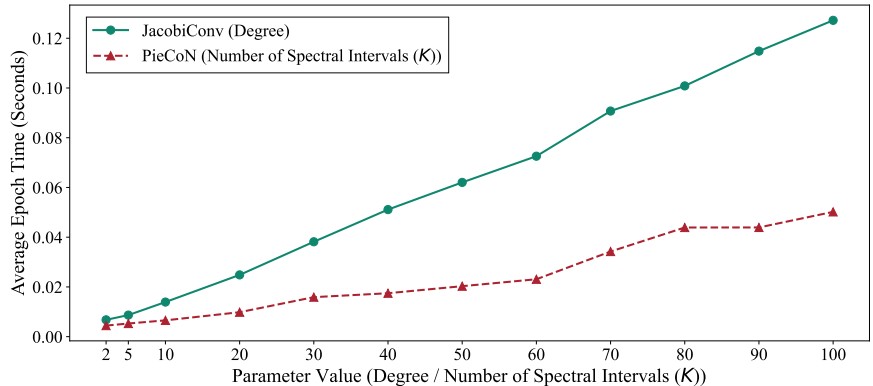

Figure 10: Computation time of PieCoN vs. JacobiConv.

Figure 10 illustrates the average epoch time (in seconds) as we increase the number of spectral intervals ($K$) in PieCoN compared to increasing the polynomial degree in JacobiConv. For a fair comparison, we disabled the polynomial part in the PieCoN implementation for this experiment.

The results show that PieCoN's constant filters require less computation time than high-degree polynomial filters. At parameter value $K = 100$, PieCoN takes approximately 0.05 seconds per epoch compared to JacobiConv's 0.13 seconds. This efficiency comes from our direct spectral interval filtering approach, which applies filters to specific eigenvalue intervals rather than performing sequential matrix multiplications needed for higher-degree polynomials.

# D  Analysis of Graph Structure and Eigenvalue Zero

The presence of eigenvalue 0 in graph spectra reveals important structural properties that many polynomial-based GNN methods overlook. Real-world datasets often exhibit high multiplicity of eigenvalue 0 (Table 1), yet methods like JacobiConv (Wang & Zhang, 2022), Bernnet (He et al., 2021), and Chebynet (Defferrard et al., 2016), which use low-degree polynomials of the normalized adjacency matrix $\hat{\boldsymbol{A}}$, do not adequately capture these properties.

**Theorem 3** (Banerjee (2008)). *Let $J$ be a graph and $H$ be a subgraph of $J$ with eigenvalue 0. Then $V(H) = p_1, p_2, \ldots, p_m \subseteq V(J)$ and $E(H) \subseteq E(J)$ consists of edges between vertices in $V(H)$. We define $J^H$ to be the graph obtained from $J$ by attaching a copy of $H$ as follows: (1) add a new set of vertices $q_1, q_2, \ldots, q_m$ to $J$, where each $q_i$ corresponds to $p_i$ in $H$; (2) add edges between vertices in $q_1, q_2, \ldots, q_m$ that mirror the edges in $H$; (3) for each $i \in 1, 2, \ldots, m$ and each vertex $r \in V(J) \setminus V(H)$, add an edge $q_i, r$ if there exists an edge $p_i, r$ in $J$.*

*Then, the graph $J^H$ has an eigenvalue 0 with an associated eigenvector $u$ that is nonzero only at the nodes $p_i$ and $q_i$. Furthermore $u_{p_i} = -u_{q_i}$.*

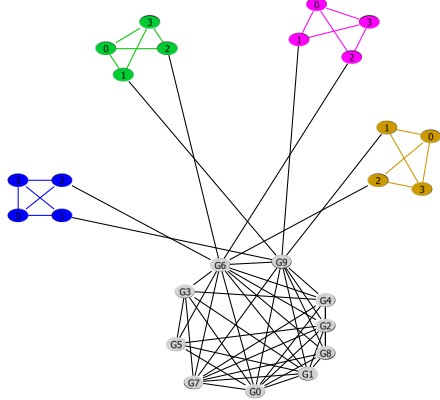

Figure 11: Simple graph with duplicate subgraphs. After adding the duplicates, the multiplicity of eigenvalue 0 increases from 0 to 3.

Theorem 3 reveals that when a graph contains duplicate substructures, it leads to eigenvalue 0 with eigenvector localized to specific node sets. This localization property is particularly relevant for node classification and community detection tasks, as nodes with similar structural roles often share the same labels (Fig. 11).

In analyzing how the negative and positive parts of $\boldsymbol{T}_k$ affect and change the structure of the graph, we consider a simple graph with duplicate subgraphs, as illustrated in Fig. 11. In this graph, nodes with the same labels are duplicates. According to Theorem 3, these duplicates create eigenvalues equal to 0 in the

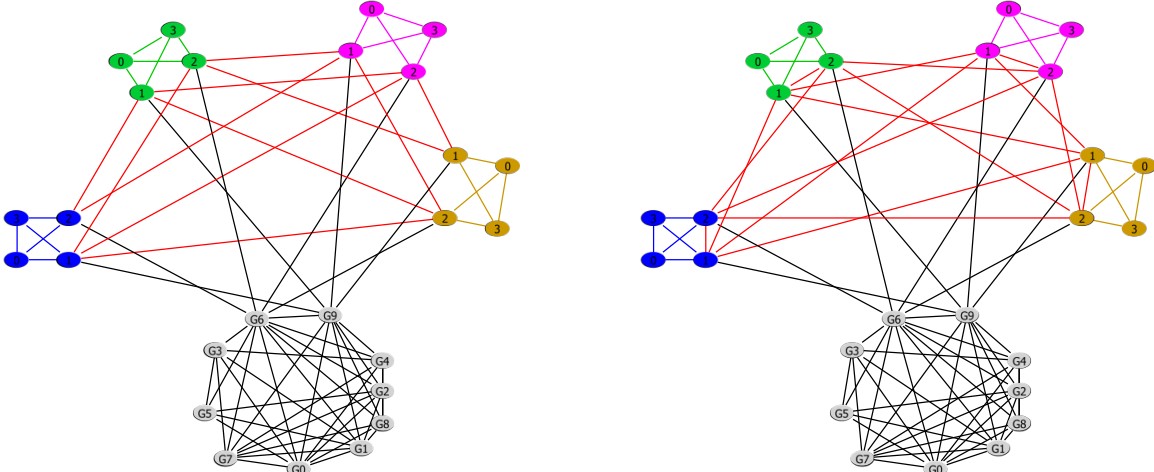

Figure 12: Simple graph with added positive edges.    Figure 13: Simple graph with added negative edges.

Table 4: Hyperparameter ranges used for optimization.

| Hyperparameter | Values |
|---|---|
| Learning Rate (`lr`) | 0.0005, 0.001, 0.005, 0.01, 0.05 |
| Weight Decay (`weight_decay`) | 0.0, 5e-5, 1e-4, 5e-4, 1e-3 |
| Feature Dropout (`feat_dropout`) | 0.0, 0.1, 0.2, 0.3, 0.4, 0.5, 0.6, 0.7, 0.8, 0.9 |
| Number of Layers (`nlayer`) | 1, 2, 3, 4, 5 |
| Hidden Dimension (`hidden_dim`) | 16, 32, 64 |
| Window size for Algorithm 1 (`window_size`) | 5, 10, 15, 20, 25, 30, 35, 40, 45, 50, 55, 60, 65, 70, 75, 80, 85, 90, 95, 100 |
| Number of spectral intervals (`num_limits`) | 0, 1, 2, 3, 4, 5, 6, 7, 8, 9, 10, 11, 12, 13, 14, 15, 16, 17, 18, 19, 20 |

eigendecomposition of the normalized Adjacency matrix. Let $\boldsymbol{U}_0$ denote the eigenvectors corresponding to the eigenvalue 0. We can decompose $\boldsymbol{R}_0 = \boldsymbol{U}_0\boldsymbol{U}_0^\top$ into its negative ($\boldsymbol{R}_0^-$) and positive parts ($\boldsymbol{R}_0^+$). Using these parts, we construct two graphs by choosing the edges with the highest score in these matrices. The graphs resulting from this process, including original and added edges, are shown in Figures 12 and 13.

From these graphs, we observe that both negative and positive edges identify connections between duplicate motifs. The negative edges also reveal connections between duplicate nodes within these duplicate motifs, highlighting their role in capturing structural similarities.

Another intuition to split $\boldsymbol{T}_k$ is that, for example, the second eigenvector provides a direction that best separates the graph into two groups while minimizing the connections cut between them. The positive and negative values show two clusters that are internally connected but separated from each other (Fiedler, 1973; Von Luxburg, 2007).

# E    Hyperparameter Optimization and Running Environment

All experiments were carried out on a Linux machine with an NVIDIA A100 GPU, Intel Xeon Gold 6230 CPU (20 cores @ 2.1GHz), and 24GB RAM. Hyperparameter tuning was performed using the Hyperopt Tree of Parzen Estimators (TPE) algorithm (Bergstra et al., 2011) with the hyperparameter ranges shown in Table 4.

The Adam optimizer was used for training with 2,000 epochs. Hyperparameters were selected to achieve the best performance on a validation set.

