# OpenReview forum: "Piecewise Constant Spectral Graph Neural Network"
_TMLR — Accepted by TMLR_

### Review · Reviewer_Dk7G · 2025-02-23

**Summary Of Contributions:**

In this article, the authors propose a design of a spectral GNN to circumvent the limitations of existing spectral GNNs that use polynomial activations. After a brief overview of the existing types of spectral GNNs, including the polynomial ones, the authors present the design of their Spectral GNN PieCON. After showing some properties of PieCON, the authors compare experimental performance of PieCON to existing GNNs on node classification tasks.

**Audience:**

Yes

**Broader Impact Concerns:**

Not applicable.

**Claims And Evidence:**

Yes

**Requested Changes:**

I would suggest the authors to:

- Give a clear mathematical definition of what a Spectral GNN is, how it differs from a standard GNN, and how PieCoN matches the definition with the appropriate choice of filters.

- Enrich the motivation of the study, for example by additional small experiments that would further support the messages of Figures 1 and 3.

- Reposition and complete the theoretical analysis of Section 5. For example, how does PieCON combines the best of both worlds (constant and polynomial) -- if it actually does.

**Strengths And Weaknesses:**

The article studies an important problem, the design of GNNs for learning problems on graphs. This problem is of practical importance as GNNs have many variants.
The article has the merit to contain extensive numerical experiments to compare their baseline to existing variants of GNNs, for node classification on various datasets.

However the article has several major weaknesses and issues as outlined below:

(i) The article does not give precise presentations of what a GNN or spectral GNN is (and a comparison thereof), in particular Section 3 is relatively poor and does not connect the rest of the paper well. A detail: Section 3 contains only one subsection (3.1). This would be useful to state it formally to clarify ideas.

(ii) The authors do not present very clearly the scientific problem addressed by the paper. For example, the choice of Figure 1 is not very informative. The figure only displays the behavior of a polynomial vs. a piecewise polynomial. The authors mention that the piece can ``focus''  on selected intervals, but this is very informal, some more explanation should be given on what this precisely means in the context of GNNs, and how it connects to some of their properties (for example, expressivity).

(iii) The theoretical analysis' content.
- Section 5.1 The Theorem 1 supposes that the polynomial has infinity norm 1 on [-1, 1]. Why is it the case? To my understanding, if this assumption is removed than the lower bound can get negative (hence uninformative) by simply scaling the coefficients of a polynomial, without increasing its degree. The authors should at least justify this assumption.
- Section 5.2: This section would be more complete if one would highlight specificities of the polynomial / constant filters. Is it true that only the piecewise/constant filters have those properties? Otherwise, it is not clear what is the theoretical gain of PieCON vs the polynomial ones. Furthermore, to my understanding, PieCON is not polynomial nor constant, it is piecewise polynomial (it has finitely many polynomial pieces).

On a minor note, the writing in general is uneven throughout the article to convey the important messages about the above points, and would gain from proofreading.

Therefore, I believe the article is not ready for publication under this form. It would benefit from major changes, and repositioning. Please see below the suggestions that would strenghten the article.

---

> ### Author Response · Authors · 2025-03-18
> **Response to Reviewer Dk7G**
>
> We thank the reviewer for their valuable feedback and constructive criticism. We address your comments below. In the paper, we marked our revisions in blue.
>
> ### **Requested Change 1:** Clearer definition of spectral GNNs vs. spatial GNNs
>
> We have expanded Section 3 to provide more precise mathematical definitions and clear distinctions between spatial and spectral GNNs
>
> ### **Requested Change 2:** Enrich the motivation of the study
>
> **Regarding Figure 1 (Filter Comparison):** We've expanded the discussion to clarify the critical advantage of our approach. As shown in this figure (Figure 1), PieCoN creates a sharp drop at eigenvalue 0, which is impossible to achieve with low-degree polynomial filters like JacobiConv. This sharp discontinuity is crucial because eigenvalue 0 often has high multiplicity in real-world graphs (as detailed in Table 1) and contains important structural information. Appendix D investigates the significance of eigenvalue 0, showing how duplicate substructures in graphs lead to this eigenvalue with localized eigenvectors. We have also added a computational comparison in Appendix C showing high interval PieCoN is more efficient than high-degree polynomial filters on each epoch.
>
> **Regarding Algorithm 1 and Figure 3:** We've provided a more thorough explanation of how our algorithm identifies significant points in the eigenvalue spectrum by detecting large gaps that represent how much eigenvalue $\lambda_i$ deviates from the statistical properties of its neighboring eigenvalues. We have added a comprehensive parameter sensitivity analysis in Appendix B (Figures 6-9) showing how window size and number of spectral intervals from Algorithm 1 affect model performance on different datasets.
>
> ### **Requested Change 3:** Reposition and complete the theoretical analysis of Section 5
>
> **Regarding polynomials having infinity norm 1:** We now assume that the filter function has norm $\lVert f\rVert_{\infty} = 1$, and we've added a new theorem (Theorem 1) that establishes error bounds for $\epsilon$-dense eigenvalues. This new theorem shows that for a filter function with  $\lVert f\rVert_{\infty} = 1$, it's natural to constrain polynomial approximations to  $\lVert p\rVert_{\infty} \leq 1$ to achieve lower approximation error.
>
> **Regarding sign and basis invariance:** Both polynomial filters and constant filters individually possess sign and basis invariance (Propositions 1-2). Some models like PP-GNN (Lingam et al. 2022) lack these properties. PieCoN inherits these properties through a linear combination of constant and polynomial filters.
>
> **Clarification on PieCoN's structure:** PieCoN is not a piecewise polynomial but rather combines:
> 1. Multiple constant filters operating on different spectral intervals.
> 2. Polynomial filters operating across the entire spectrum.
>
> This design provides both sharp responses at crucial eigenvalues (constant filters) and smooth transitions for gradual spectral changes (polynomial filters). The ablation study in Table 3 confirms this complementary approach outperforms either method alone.
>
> ### **Minor Changes:**
> - We have thoroughly proofread the manuscript to improve clarity and consistency in writing.
> - We moved detailed proofs to the appendix to improve readability.
>
> We are grateful for the thorough review, which has helped us address important aspects of our method and strengthen the paper.

---

> > ### Comment · Reviewer_Dk7G · 2025-04-02
> > **Follow up**
> >
> > I appreciate the authors' answers that clarified some points, especially the clearer definition of spectral GNNs vs. spatial GNNs
> > However, I still have some concerns with the new version:
> >
> > - In Section 5. Choosing a specific family of filters (those with infinity norm at most one) is relevant to show some limitations of the other models, using lower bounds on the error. The obtained lower bound from Theorem 1 justifies the assumption on the norm of the polynomial chosen (otherwise the error gets worse). However, in Theorem 2, there are two things that are problematic:
> > a) One supposes that there is a ``discontinuous'' jump of magnitude between consecutive eigenvalues. This seems difficult to be compatible with the framework of Theorem 1 where the eigenvalues are \epsilon-dense (or $h < \epsilon $ and the lower-bound becomes likely uninformative because negative)
> > b) If $h$ is the amplitude of the jump between $\lambda_R  $ and $\lambda_{R+1}$, then shouldn't the bound (12) simplify to $\geq (h(1 - d^2))$? Again, if d > 1, the bound becomes uninformative.
> >
> > - In their answer, the authors say that PieCON is not piecewise polynomial. However, they also say that PieCON is a linear combination of constant and polynomial filters, which is piecewise polynomial. However, a linear combination of constant and polynomial functions is piecewise polynomial (according to the following definition: a function is piecewise-polynomial iff there exists finitely many regions of the domain considered where in each region, it coincides with some polynomial).
> >
> > - The computational cost of the overall method is at least \Omega(n^3). Does it constitute a practical limitation? Can you elaborate on the size of the graphs that can be handled in practice with the spectral GNN methods?

---

> > > ### Author Response · Authors · 2025-04-03
> > >
> > > We thank the reviewer for their insightful follow-up questions. We address each of the concerns below:
> > > 1. In Theorem 2, the assumption of a "discontinuous" jump of magnitude $h$ refers to the function difference: $|f(\lambda_{R+1}) - f(\lambda_{R})| = h∣$ rather than the eigenvalue gap $|\lambda_{R+1} - \lambda_R| = h$. This theorem complements Theorem 1, as the bound in (12) tends to $h$ when $\epsilon \to 0$ ($\epsilon>|\lambda_{R+1} - \lambda_R|$), ensuring consistency between the two results.
> > > 2. The reviewer is correct that our method is technically a piecewise polynomial. We wanted to clarify that we don't construct separate polynomials for each interval – instead, we use constant functions within intervals combined with global polynomials.
> > > 3. As stated in Section 4.5, our model's precomputation requires $O(n^3)$ time (eigendecomposition + filter construction ($O(n^3 + 2Kn^2 \log (m)) = O(n^3)$)). This preprocessing step occurs only once and has become practical with modern GPUs - for example, it takes just 8 minutes for the Amazon-Ratings dataset (25,000 nodes) on an NVIDIA A100 GPU. After preprocessing, training and inference complexity becomes $O((K + P)md)$, competitive with other spectral methods.\
> > > Regarding the practical size limitations: Our approach is designed for graphs of reasonable size (up to tens of thousands of nodes). For graphs of this scale, the eigendecomposition is a worthwhile one-time preprocessing cost given the expressivity benefits.\
> > > While spectral methods like ChebNet and JacobiConv avoid explicit eigendecomposition, they still face challenges with higher-degree polynomials (needed for better approximation), as the recursive formulation to compute $A^k X$ becomes computationally intensive in practice. They will also struggle when scaling to hundreds of thousands of nodes due to memory limitations for storing large sparse matrices and intermediate results. In practice, all these spectral methods (including ours) are typically evaluated on graphs with thousands to tens of thousands of nodes. \
> > > We want to clarify that our paper does not address the problem of scaling GNNs to massive graphs with millions or billions of nodes. This is a separate research area beyond our scope. For such massive graphs, even standard MPNNs require specialized techniques like graph sampling or partitioning, which we do not explore in this work.

---

> > > > ### Comment · Reviewer_Dk7G · 2025-04-04
> > > > **Answer to 3 points**
> > > >
> > > > Thanks for your reply.
> > > >
> > > > I encourage the authors to include discussion in points 2 and 3 in the paper.
> > > >
> > > > Regarding point 1, I thank the authors for the clarification. One additional comment: if the function $f$ is supposed Lipshitz, then
> > > > $\lvert f(\lambda_{R+1}) - f(\lambda_{R}) \rvert = h$ for some $R$ implies, $\lvert \lambda_{R+1} - \lambda_{R} \rvert \geq \frac{h}{L}$. So now the regime $\lvert \lambda_{R+1} - \lambda_{R} \rvert << \frac{h}{d^2}$ becomes $L >> d^{2}$.
> > > > In other terms, if the Lipshitz constant is large compared to $d^{2}$, then the error is at least of order $h$.
> > > >
> > > > I have no further question.

---

### Review · Reviewer_vcVp · 2025-02-24

**Summary Of Contributions:**

The authors provide an error bound for spectral polynomial approximation and show that using piecewise constant filters can remove this error. This motivates combining constant spectral filters with polynomial filters to increase the range of spectral properties that can be effectively learned without using computationally expensive high degree polynomials. The authors show that this combination performs well on a range of homophilic and heterophilic tasks.

**Audience:**

Yes

**Broader Impact Concerns:**

none required

**Claims And Evidence:**

Yes

**Requested Changes:**

My requested changes go some way to reduce the weaknesses discussed in the previous section.

- I strongly encourage timing experiments of the approach with baseline comparisons to different degrees of polynomial filters (weakness 1).
- Results should be shown in order to better understand the effects of Hyperparameters in the Thresholding Algorithm (weakness 2).
- The authors should be clearer on the benefits of the approach with regards to the multiplicity of eigenvalues. The quoted multiplicity in Table 1 should be revisited and explained in more detail (weakness 3).
- PP-GNN could be added as a baseline in Table 2.
- The bold numbers should reflect statistical significance in the results tables.

**Strengths And Weaknesses:**

Strengths

- The error bound analysis clearly motivates the approach, with low degree polynomial filters unable to account for jump discontinuity between eigenvalues which are close.

- The proposed method is shown to perform well on a range of tasks, providing some evidence for the efficacy of the approach.

- Whilst polynomial filters are sign and basis invariant, the authors highlight that their full method is also sign and basis invariant. This demonstrates a more detailed evaluation of their approach.

Major Weaknesses

- The error in the bound of Section 5.1 can be reduced using a high degree polynomial. The authors suggest that the limitation of this is that they are computational expensive. However, the method proposed by the authors also increases the computation as outlined in 4.5. Given that the method is motivated because of the computational limitation of high degree polynomials, more work should be done to compare their complexity and the relative benefits of your approach. A timing comparison of standard polynomial filter, high degree polynomial filter and your method would help a lot in this regard.

- The authors highlight that “the model’s performance is highly dependent on how we partition the eigenvalue intervals”. Adding the results which show this are important to provide a better understanding of the method. The method provides additional hyperparameters within the thresholding algorithm so showing how robust the results are over changes to these parameters and what the optimal values are, will better highlight the practical value of the approach.

- At multiple points in the paper, the authors claim that their method would improve for cases where there exists eigenvalues with large multiplicities. This is not clear from the paper as you have the same filter output for a given eigenvalue (shown in Figure 1). From the bound and the analysis, it seems the method would help when eigenvalues are closely spaced but not when they are identical.

Minor Weaknesses

- The bold text used in the experimental results is quite misleading. All top results that are statistically equivalent should be highlighted in bold. For example, in the ablation on Chameleon in Table 3 using positive+negative without the polynomial looks not significantly worse than positive+negative+polynomial. Both should really be in bold. This should be done for all statistically equivalent results.

- PP-GNN is mentioned in the paper to be the most similar method [Lingam et al, 2022]. However, I don’t see any performance comparison. This should be done to show why your method is better than only using two frequency bands.

- You say that “real-world graphs where certain eigenvalues like zero have **large** multiplicities”. This multiplicity is given by the number of connected components in the graph, so I am surprised this would be large in many practical scenarios. In Table 1, you claim to list this multiplicity (Lamba(0)/Nodes), but this looks like it should be labelled (Lamba(1)/Nodes). For example, on Cora (Lambda(1)/Nodes) is 0.11, as in the table, but (Lambda(0)/Nodes) is much less.

[Lingam et al, 2022] Vijay Lingam et al. A piece-wise polynomial filtering approach for graph neural networks. In ICLR Workshop
on Geometrical and Topological Representation Learning, 2022.

---

> ### Author Response · Authors · 2025-03-18
> **Response to Reviewer vcVp**
>
> We thank the reviewer for the thoughtful comments and suggestions. We address your comments below. In the paper, we marked our revisions in blue.
>
> ### **Requested Change 1:** Timing experiments
>
> We have added a computational comparison in Appendix C showing high interval PieCoN is more efficient than high-degree polynomial filters on each epoch. Furthermore, multiple works have demonstrated that increasing polynomial degree beyond certain thresholds leads to performance plateaus or degradation:
>
> [1] Zhang et al., "Bernnet: Learning arbitrary graph spectral filters via Bernstein approximation," NeurIPS 2021. (Figure 5)
>
> [2] Kipf et al., "Semi-supervised classification with graph convolutional networks," ICLR 2017. (Figure 5)
>
> [3] Wu et al., "Simplifying graph convolutional networks," ICML 2019. (Figure 4)
>
> [4] Alon et al., "On the bottleneck of graph neural networks and its practical implications," ICLR 2021. (Figure 3)
>
> ### **Requested Change 2:** Effect of Hyperparameters in Thresholding Algorithm
>
> We have added an ablation study on the impact of window size $w$ and the number of partitions K in Appendix B.
>
> ### **Requested Change 3:** Clarification on eigenvalue multiplicity
>
> We have added a footnote in the Introduction to clarify that we are examining the multiplicity of eigenvalue 0 in the normalized adjacency matrix, which corresponds to eigenvalue 1 in the normalized Laplacian matrix. In Subsection 6.1 Datasets, we have also added a sentence explaining the last column of Table 1.
>
> ### **Requested Change 4:** PP-GNN baseline
>
> Following the reviewer's suggestion, we've added PP-GNN as a baseline. Experimental results show that PieCoN outperforms PP-GNN across all datasets, demonstrating the effectiveness of our design choices. To view all the differences between our model and PP-GNN please see our response to the Reviewer VxFY (Requested Change 1).
>
> ### **Requested Change 5:** Statistical significance in results tables
>
> We implemented t-tests for all comparisons, marking significant differences with asterisks (* for p < 0.05, *** for p < 0.001) while keeping bold for the highest values to be consistent with common practices in the literature (Tables 2 and 3).
>
> ### **Major Weakness 3:** Problem of polynomial filters when dealing with large multiplicity
>
> Low-degree polynomial filters cannot assign significantly different weights to eigenvalues with high multiplicity compared to their neighbors due to their inherent smoothness constraints. This prevents proper emphasis on these structurally important eigenvalues. Our constant filters enable distinct treatment of high-multiplicity eigenvalues independently from neighboring eigenvalues, as shown in Figure 1 where there is a sharp drop in eigenvalue zero.
>
> We thank the reviewer for their valuable suggestions that have significantly enhanced the rigor and quality of our manuscript.

---

### Review · Reviewer_VxFY · 2025-02-25

**Summary Of Contributions:**

The paper proposes PieCoN, a spectral GNN which includes constant spectral filters alongside polynomial filters to capture both sharp and gradual changes in the graph’s spectral properties. Theoretically, the authors provide an error bound for spectral polynomial approximation, showing that commonly used polynomial filters (which have small degree) are limited when approximating functions with sharp changes. Empirically, the proposed method outperforms existing polynomial-filters based spectral GNNs and spatial GNNs on 5 out of 7 datasets.

**Audience:**

Yes

**Claims And Evidence:**

Yes

**Requested Changes:**

- Please discuss the difference with PP-GNN (Lingam et al. 2022) in more detail.
- Please explain what is the problem of polynomial filters when dealing with large multiplicity (said in the introduction). Indeed, while it is clear that when using low degree polynomials the change between closely spaced eigenvalues cannot vary significantly, I do not see how it connects to large multiplicity (where the eigenvalues are not closely spaced, but identical).
- Can you explain why we need the polynomial filters instead of relying on constant filters only? I think this is especially relevant because we could impose additional constraints to maintain continuity between contiguous intervals, as in PP-GNN.
- Algorithm 1, which is used to partition the graph spectrum, should be explained and not just referenced in the text. I think the algorithm just identifies eigenvalues that deviate significantly from the surrounding values by comparing them to local mean values, but please expand.
- **The paper needs to include the empirical comparison with the related work also targeting the limited flexibility of polynomial filtering**, such as  DFS (Guo et al., 2023), NFGNN (Zheng et al., 2023) and PP-GNN (Lingam et al. 2022). Indeed, as these works are presented in the related work section they should serve as a direct baseline.
- Related to the above, please add the missing datasets that are present in DFS (Wisconsin Cornell Texas Twitch-DE Pubmed Computers Photo), and explain why the reported numbers on the datasets that are present are different from those reported in DFS.
- What is the impact of the window size W and the impact of the number of partitions K? Please add an ablation study.


Additional minor changes:
1. Please move the proofs to the appendix and provide only an high level proof overview in the main paper.
2. I think using W for both window size and for the weights is confusing. Maybe you can denote the window size by w instead of W.

**Strengths And Weaknesses:**

### Strengths
1. The paper is clear, and despite many concepts that are introduced, it is easy to follow.
2. The solution idea is simple, even though the specific algorithm proposed for partitioning the graph spectrum into different intervals is not well motivated in the writing.


### Weaknesses
1. The contribution of the paper appears limited as the method is very similar to PP-GNN (Lingam et al. 2022). Indeed, PP-GNN fits a low degree polynomial in each interval, while PieCoN sums a polynomial to a filter that is constant in each interval. Please clarify the implications of this difference.
1. The paper claims that polynomial filters may not *``fully identify graph's spectral properties''* (e.g., in the abstract), without actually introducing what this means in practice, nor discussing how including constant filters fixes this problem. From the theoretical analysis, I believe that the authors mean that polynomial filters cannot approximate functions with sharp changes if the polynomial degree is small, while constant filters can. Being more concrete on this point can significantly improve the paper.
2. The theoretical results are not very surprising, especially Proposition 1 (which follows directly from Lim et al., 2023) which I think can be presented in words.

---

> ### Author Response · Authors · 2025-03-18
> **Response to Reviewer VxFY**
>
> We thank the reviewer for their thoughtful and constructive feedback. We address your comments below. In the paper, we marked our revisions in blue.
>
> ### **Requested Change 1:** PP-GNN Comparison
>
> While both approaches use piecewise spectral filtering, PieCoN differs from PP-GNN in several important ways:
>
> 1. Adaptive vs. fixed partitioning: PP-GNN employs a fixed two-part partitioning scheme dividing the spectrum into top-k and bottom-k eigenvalues. In contrast, PieCoN implements an adaptive $K$-part partitioning through Algorithm 1, which identifies significant points in the eigenvalue spectrum. This allows PieCoN to better adapt to diverse graph structures rather than using predetermined cutoffs.
> 2. Filter design: PP-GNN fits low-degree polynomials in each interval, while PieCoN uses constant filters on intervals combined with global polynomial filters.
> 3. Discontinuity as a feature: While PP-GNN imposes continuity constraints between intervals, we intentionally allow discontinuities in the filter response. This is a key feature of our approach, not a limitation. Just because two eigenvalues are close in the spectrum does not mean their weights in the filtering function should be similar.
> 4. Positive/negative decomposition: Our approach uniquely decomposes constant filters into positive and negative parts (as shown in Eq. 10). This decomposition helps identify structural similarities in graphs, particularly for capturing connections between duplicate motifs (as illustrated in Appendix D).
> 5. Invariance properties: As demonstrated in Propositions 1 and 2, our model maintains invariance to sign flips and basis changes in the eigenvectors, which is crucial for model stability and generalization. PP-GNN's approach of selecting top-k and bottom-k eigenvalues might not preserve this invariance property depending on the graph.
> 6. Superior performance: Experimental results (Table 2) show that PieCoN outperforms PP-GNN across all datasets, demonstrating the effectiveness of our design choices.
> We have expanded our related work section to better highlight these differences.
>
> ### **Requested Change 2:** Problem of polynomial filters when dealing with large multiplicity
>
> Low-degree polynomial filters cannot assign significantly different weights to eigenvalues with high multiplicity compared to their neighbors due to their inherent smoothness constraints. This prevents proper emphasis on these structurally important eigenvalues. Our constant filters enable distinct treatment of high-multiplicity eigenvalues independently from neighboring eigenvalues, as shown in Figure 1 where there is a sharp drop in eigenvalue 0.
>
> ### **Requested Change 3:** Need for polynomial filters
>
> Combining constant and polynomial filters provides complementary benefits:
> Polynomial filters capture smooth spectral transitions and multi-hop neighborhood information through powers of the adjacency matrix, where $A^p$ aggregates $p$-hop neighborhood data.
> Constant filters enable sharp responses at critical eigenvalues, which is particularly beneficial for heterophilic graphs with high eigenvalue multiplicity.
> Our ablation study (Table 3) confirms that this combination consistently outperforms either approach alone across most datasets.
>
> ### **Requested Change 4:** Algorithm 1 explanation
>
> We have expanded the description of Algorithm 1 in the paper.
>
> ### **Requested Change 5 and 6:** Empirical comparison and additional datasets
>
> We have enhanced our experimental evaluation by:
> Adding comparisons with DSF-Jacobi-R and PP-GNN in Table 3, showing that PieCoN outperforms these methods across most datasets.
> Including Texas and Pubmed datasets (Amazon-Photo was already included).
> Implementing statistical significance testing (t-tests) to validate performance differences.
> The differences in reported numbers compared to DFS are due to our use of random 60%/20%/20% train/validation/test splits across 10 seeds, rather than using pre-defined splits.
>
> ### **Requested Change 7:** Algorithm 1 parameter ablation study
>
> We have added an ablation study on the impact of window size $w$ and the number of partitions $K$ in Appendix B.
>
> ### **Minor Changes:**
>
> We have moved the detailed proofs to the appendix to improve readability and have changed the notation for window size from $W$ to $w$ to avoid confusion with the weight matrices $W^1$ and $W^2$. We’ve also added a comment for Proposition 1 that directly follows from Lim et al., 2023.
> We appreciate the constructive feedback, which has helped us improve the clarity and comprehensiveness of our paper.

---

> > ### Comment · Reviewer_VxFY · 2025-03-27
> > **Follow up**
> >
> > I appreciate the authors' response and have two minor follow-up questions:
> >
> > 1. Does the issue with polynomial filters handling large multiplicity arise only when neighboring eigenvalues do not also have large multiplicity?
> >
> > 2. Why did you choose not to use the predefined splits available in the datasets?

---

> > > ### Author Response · Authors · 2025-03-28
> > >
> > > We thank the reviewer for their thoughtful follow-up questions.
> > >
> > > 1. Not necessarily. Even when neighboring eigenvalues have high multiplicity, they often represent distinct structural properties requiring different weighting. Additionally, this scenario (adjacent eigenvalues all having high multiplicity) rarely occurs in real-world graphs, including all datasets in our study. Our method provides flexibility to differentiate between these eigenvalue clusters through sharp transitions in the filter response.
> > >
> > > 2. Predefined splits can lead to models tuned specifically for those partitions rather than demonstrating general performance. As shown by Shchur et al. [1], using the same train/validation/test splits might prevent a fair comparison across different architectures, creating the misconception that some models perform better than others, when this advantage may simply come from favorably fitting those specific splits. We followed the methodology established in recent spectral GNN works (JacobiConv [2], OptBasisGnn [3], Specformer [4]]) by using random 60/20/20 splits across 10 seeds, to avoid this issue.
> > >
> > > [1] Shchur et al., Pitfalls of Graph Neural Network Evaluation, NeurIPS 2018.
> > >
> > > [2] Wang et al., How powerful are spectral graph neural networks, ICML 2022.
> > >
> > > [3] Guo et al., Graph neural networks with learnable and optimal polynomial bases, ICML 2023.
> > >
> > > [4] Bo et al., Specformer: Spectral graph neural networks meet transformers, ICLR 2023.

---

### Decision · Action_Editor_d57K · 2025-04-05

**Recommendation:** Accept with minor revision

**Comment:**

Please carefully take into account any residual discussion with the Reviewers -- especially Reviewer Dk7G -- and incorporate any and all relevant takeaways into the camera-ready version. Once this is done, the paper should be ready for publication in TMLR.

**Audience:**

Yes -- spectral filtering is of significant interest to a sizable proportion of the graph representation learning community, and the results have been judged to be of interest in this domain.

**Claims And Evidence:**

Yes -- after the rebuttal revision, the paper meets the technical soundness bar of TMLR in principle (though some revisions may yet be needed).

---

> ### Author Response · Authors · 2025-04-29
>
> Thank you for your thoughtful evaluation of our paper. We appreciate your decision to accept our work with minor revision.
>
> 1. Reviewed and incorporated the remaining discussion points from all reviewers, with particular attention to Reviewer Dk7G's feedback.
> 2. We incorporated all relevant feedback into the camera-ready version.
>
> The manuscript has been thoroughly revised according to these recommendations, and we believe it is now ready for publication in TMLR.